# Interaction between coxsackievirus B3 infection and α-synuclein in models of Parkinson's disease

Soo Jin Park[1,2,3,4], Uram Jin[1,2,4,5], Sang Myun Park[1,2,4]*

1 Department of Pharmacology, Ajou University School of Medicine, Suwon, Korea, 2 Center for Convergence Research of Neurological Disorders, Ajou University School of Medicine, Suwon, Korea, 3 Department of Thoracic and Cardiovascular Surgery, Ajou University School of Medicine, Suwon, Korea, 4 Department of Biomedical Sciences, Ajou University School of Medicine, Suwon, Korea, 5 Department of Cardiology, Ajou University School of Medicine, Suwon, Korea

* sangmyun@ajou.ac.kr

**Data Availability Statement:** All relevant data are within the manuscript and its Supporting Information files.

**Funding:** This research was supported by the National Research Foundation of Korea (NRF)

## Abstract

Parkinson's disease (PD) is one of the most common neurodegenerative diseases. PD is pathologically characterized by the death of midbrain dopaminergic neurons and the accumulation of intracellular protein inclusions called Lewy bodies or Lewy neurites. The major component of Lewy bodies is α-synuclein (α-syn). Prion-like propagation of α-syn has emerged as a novel mechanism in the progression of PD. This mechanism has been investigated to reveal factors that initiate Lewy pathology with the aim of preventing further progression of PD. Here, we demonstrate that coxsackievirus B3 (CVB3) infection can induce α-syn-associated inclusion body formation in neurons which might act as a trigger for PD. The inclusion bodies contained clustered organelles, including damaged mitochondria with α-syn fibrils. α-Syn overexpression accelerated inclusion body formation and induced more concentric inclusion bodies. In CVB3-infected mice brains, α-syn aggregates were observed in the cell body of midbrain neurons. Additionally, α-syn overexpression favored CVB3 replication and related cytotoxicity. α-Syn transgenic mice had a low survival rate, enhanced CVB3 replication, and exhibited neuronal cell death, including that of dopaminergic neurons in the substantia nigra. These results may be attributed to distinct autophagy-related pathways engaged by CVB3 and α-syn. This study elucidated the mechanism of Lewy body formation and the pathogenesis of PD associated with CVB3 infection.

## Author summary

Prion-like propagation of α-syn has emerged as a novel mechanism involved in the progression of Parkinson's disease (PD). This process has been extensively investigated to identify the factors that initiate Lewy pathology to prevent further progression of PD. Nevertheless, initial triggers of Lewy body (LB) formation leading to the acceleration of the process still remain elusive. Infection is increasingly recognized as a risk factor for PD. In particular, several viruses have been reported to be associated with both acute and chronic

grants funded by the Korean government (Ministry of Science and ICT and Ministry of Education) (grant No. NRF-2017R1E1A1A01073713 to SMP, NRF-2019R1A5A2026045 to SMP and M-2021A040300173 to UJ). The funders had no role in study design, data collection and analysis, decision to publish, or preparation of the manuscript.

**Competing interests:** The authors have declared that no competing interests exist.

parkinsonism. It has been proposed that peripheral infections including viral infections accompanying inflammation may trigger PD. In the present study, we explored whether coxsackievirus B3 (CVB3) interacts with α-syn to induce aggregation and further Lewy body formation, thereby acting as a trigger and whether α-syn affects the replication of coxsackievirus. It is important to identify the factors that initiate Lewy pathology to understand the pathogenesis of PD. Our findings clarify the mechanism of LB formation and the pathogenesis of PD associated with CVB3 infection.

## Introduction

Parkinson's disease (PD) is one of the most common neurodegenerative diseases. PD is pathologically characterized by the death of midbrain dopaminergic neurons and the accumulation of intracellular protein inclusions termed Lewy bodies (LBs) or Lewy neurites (LNs) [1, 2]. The major component of these inclusions is α-synuclein (α-syn) [3]. Protein inclusions with α-syn aggregates have also been observed in other neurodegenerative disorders, such as multiple system atrophy and dementia with Lewy bodies, which are collectively referred to as α-synucleinopathies [4]. Multiplications and missense mutations of the α-syn gene have been identified in patients with early onset familial PD [5]. Furthermore, genome-wide association studies have demonstrated a strong association between α-syn gene and sporadic PD [6, 7], suggesting a major role of α-syn in the pathogenesis of PD.

Lewy pathology first appears in the olfactory bulbs and dorsal motor nucleus of the vagus nerve, which is connected to the enteric nervous system. The pathology progressively involves more regions of the nervous system and subsequently the cortical areas as the disease advances [8]. This pathology seems to be occur prior to the appearance of motor symptoms in PD and may be associated with gastrointestinal and olfactory dysfunctions, which are frequently observed in the prodromal phase of PD [9]. Substantial *in vitro* and *in vivo* experimental evidence has implicated prion-like propagation of α-syn as a novel mechanism in the progression of PD [10–12]. Targeting this mechanism could enable the development of disease-modifying therapies for patients with PD. However, the initial triggers of LB formation leading to acceleration of the process remain elusive.

Viral infection is increasingly being recognized as a risk factor for PD. A number of viruses have been associated with both acute and chronic parkinsonism. These viruses include influenza virus, coxsackievirus, Japanese encephalitis B virus, western equine encephalitis virus, and herpes virus [13]. It has been proposed that peripheral infections, including viral infections accompanying inflammation, may trigger PD [14].

Coxsackievirus is a single-stranded RNA virus belonging to the *Picornaviridae* family of viruses in the genus *Enterovirus* [15]. More than 90% of the coxsackievirus infections are asymptomatic. Clinically, infants or young adults are easily infected with this virus, and a few develop severe myocarditis [16] or meningitis [17]. Persistent coxsackievirus infection is also associated with chronic myocarditis, dilated cardiomyopathy [18], and type I diabetes [19].

A recent report described virus-like particles and enterovirus antigen in the brainstem neurons of PD [20]. This finding prompted the speculation that enterovirus infection in PD may act as a seed for the aggregation of α-syn in addition to the direct cytopathic effect of viral infection in neurons. In addition, α-syn inhibits West Nile virus (WNV) infection by acting as a viral restriction factor [21], suggesting that α-syn expression may affect viral infection in the central nervous system (CNS).

In the present study, we explored whether coxsackievirus B3 (CVB3) interacts with α-syn to induce aggregation and further LB formation, and whether α-syn affects the replication of coxsackievirus.

## Results

### CVB3 infection regulates α-syn expression in neurons

To explore whether CVB3 affects α-syn, we infected differentiated SH-SY5Y cells (dSH-SY5Y cells) with CVB3 (MOI 0.25) for 24 h. CVB3 VP1 colocalized with α-syn and the intensity of α-syn in infected cells was increased. Interestingly, we observed that CVB3 infection induced the formation of large aggregates of α-syn that completely filled the cytoplasm and pushed the nucleus aside, creating a half-moon appearance. This appearance was more pronounced in dSH-SY5Y cells overexpressing α-syn (Fig 1A). It was not due to the cross-reactivity of α-syn antibody with CVB3 (S1A Fig) and these inclusions was eosionophlic (S1B Fig). In primary cortical neurons, similar colocalization of VP1 with α-syn was observed (Fig 1B). In contrast, the mRNA levels of α-syn were decreased in dSH-SY5Y cells and primary cortical neurons infected with CVB3 (Fig 1C). No significant cytotoxicity was observed upon CVB3 infection (S1C Fig), suggesting that the decrease in α-syn mRNA level might not be due to cytotoxicity. Infection of cells with many viruses results in inhibition of transcription or translation of host cell mRNAs, termed as host shutoff [22, 23]. The levels of several mRNAs, including β-actin, histone H3, and polr2, which are known to be associated with the shutoff phenomenon [24–26], were not altered upon CVB3 infection. Moreover, the levels of PD-associated genes, such as DJ-1, PINK1, and parkin were not altered upon CVB3 infection (S1D Fig), suggesting that the decrease in α-syn mRNA level might also not be due to the host shutoff phenomenon. Analysis of an open source database (GSE 19496) also showed that the mRNA levels of α-syn in CVB3-infected mouse heart were decreased compared to those in the control (S1E Fig). Western blot showed that endogenous α-syn expression was decreased. Interestingly, the expression of ectopically overexpressed α-syn was also decreased. This phenomenon was likely to be more severe with the increase in the viral titer (Fig 1D). In addition, when we intraperito-neally infected WT mice with CVB3, decreased levels of α-syn mRNA and protein were observed in the brain (Fig 1E and 1F). Given that CVB3 did not infect all the cells, these results led us to speculate that α-syn may be regulated differently in CVB3-infected cells and neigh-boring cells. To explore this, we compared α-syn levels in non-infected and infected condi-tions. Increased α-syn levels were observed in CVB3-infected cells and the levels were markedly decreased in the cells near the CVB3-infected cells (Fig 1G). Similar findings were observed in primary cortical neurons (Fig 1H), suggesting that α-syn was regulated differently in CVB3-infected cells and neighboring cells. When cells were treated with poly IC, an artifi-cial analog to mimic RNA viral infection [27], α-syn expression was found to be increased at both the mRNA and protein levels, whereas α-syn aggregates were not observed (Fig 1I–1K). These findings suggested that the observations were specific to CVB3 infection. Analysis of another open source data (GSE 7621) also revealed decreased levels of α-syn mRNA in the brains of patients with PD compared with those in normal individuals (S1F Fig). These results suggested that CVB3 infection induced large cytosolic aggregates that colocalized with α-syn, and that the expression of α-syn was differentially regulated in infected cells and neighboring cells.

### CVB3 infection induces LB-like inclusion body formation in neurons

CVB3 forms very large autophagy-related structures termed megaphagosomes in murine pan-creatic acinar cells, the structure of which represents a viral replication complexes [28]. To

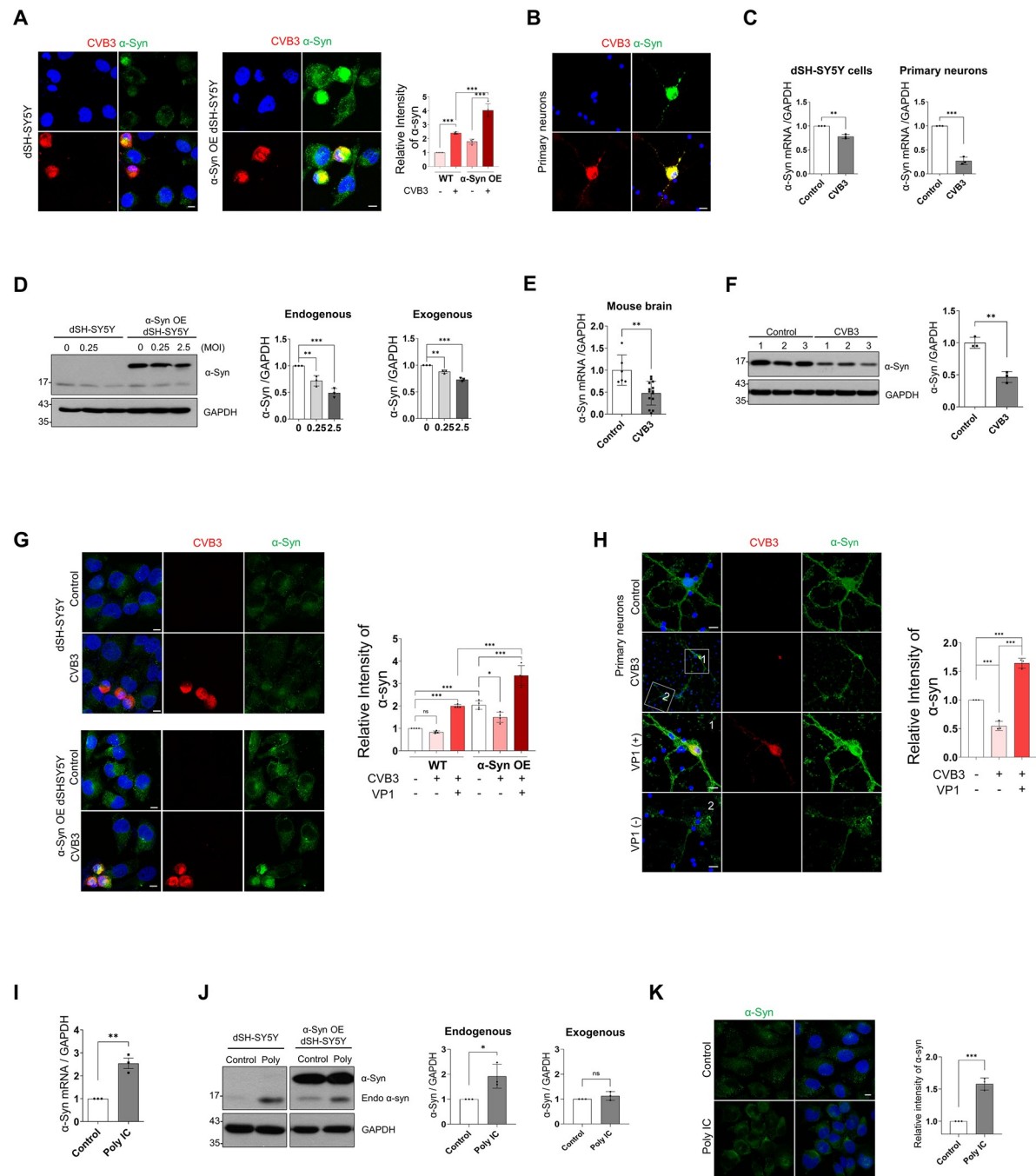

**Fig 1. CVB3 infection regulates α-syn expression in neurons.** (A) Immunocytochemistry (ICC) images of WT and α-syn OE dSH-SY5Y cells infected with CVB3 (MOI 0.25) (red) for 24 h. The intensity of α-syn (green) was analyzed. Values are derived from four independent experiments (n = 4). *** P < 0.001, one-way ANOVA test with Tukey's multiple comparison test. (B) ICC images of mouse primary cortical neurons infected with CVB3 (MOI 5) for 24 h. Scale bar indicates 10 μm. Blue indicates Hoechst. (C) The relative expression levels of α-syn mRNA between control, CVB3-infected dSH-SY5Y cells (MOI 0.25) and mouse primary cortical neurons (MOI 5) infected with CVB3 for 24 h. Values are derived from three independent experiments (n = 3). *** P < 0.001, ** P < 0.01, unpaired t-test. (D) WT and α-syn OE dSH-SY5Y cells were infected with the indicated MOIs of CVB3 for 24 h. Western blotting was performed with the indicated antibodies and protein levels were quantified by densitometry. Values are derived from three independent experiments (n = 3). *** P < 0.001, ** P < 0.01, one-way ANOVA test with Tukey's multiple comparison test. (E) The relative levels of α-syn mRNA expression between control (n = 6) and CVB3 infected mice brain hemispheres (n = 14) after 4 days of intraperitoneal (IP) injection *with* $1.0 \times 10^6$ plaque forming units (PFUs) of CVB3 in 100 μl of PBS. ** P < 0.01, unpaired t-test. (F) The relative levels of α-syn expression between control (n = 3) and CVB3 infected mice brain hemispheres (n = 3) after 7 days IP injection with $1.0 \times 10^6$ PFUs of CVB3. Western blotting was performed with the indicated

antibodies and protein levels were quantified by densitometry. ** P < 0.01, unpaired t-test. (G and H). ICC images of control and CVB3 infected WT, α-syn OE dSH-SY5Y cells (G) and mouse primary cortical neurons (H), which were infected with CVB3 for 24 h. α-Syn (green) intensity of indicated condition was analyzed. Values are derived from three (G) or four (H) independent experiments (n = 3 or 4). Scale bar indicates 10 μm. Blue indicates Hoechst. * P < 0.05, *** P < 0.001, one-way ANOVA test with Tukey's multiple comparison test. (I) The relative expression levels of α-syn mRNA between control and 1 μg/ml poly IC-transfected dSH-SY5Y cells for 4 h. Values are derived from three independent experiments (n = 3). ** P < 0.01, unpaired t-test. (J) WT and α-syn OE dSH-SY5Y cells transfected with 1 μg/ml poly IC for 4 h. Western blotting was performed with the indicated antibodies and protein levels were quantified by densitometry. Values are derived from three independent experiments (n = 3). * P < 0.05, unpaired t-test. (K) ICC image of control and 1 μg/ml poly IC-transfected dSH-SY5Y cells for 24 h. The intensity of α-syn (green) was analyzed. Values are derived from three independent experiments (n = 3). Scale bar indicates 20 μm. Blue indicates Hoechst. *** P < 0.001, unpaired t-test.

investigate whether these large aggregates colocalized with α-syn in more detail, we stained the cells for microtubule-associated protein 1A/1B light chain 3B (LC3), a marker for autophagosomes [29]. These structures completely colocalized with LC3 in dSH-SY5Y cells, α-syn over-expressing (OE) dSH-SY5Y cells and primary cortical neurons (Fig 2A). These structures also co-localized with pSer129 α-syn in α-syn OE dSH-SY5Y cells and primary cortical neurons (Fig 2B). The colocalization of these aggregates with ubiquitin, another marker for LBs [30], was more clearly observed in α-syn OE dSH-SY5Y cells, suggesting that these structures may be LB-like inclusions (Fig 2C). Upon infection with enterovirus71 (EV71), another virus of the *Picornaviridae* family, dSH-SY5Y cells formed smaller LC3-positive aggregates. However, they did not colocalize with α-syn (S2 Fig), suggesting that the formation of LB-like inclusion bodies containing α-syn was CVB3-specific. Next, we examined these structures by transmission electron microscopy (TEM). In the absence of CVB3 infection, intracellular organelles were dispersed throughout the cytoplasm in dSH-SY5Y cells and α-syn OE dSH-SY5Y cells, whereas the organelles of virus-infected cells were accumulated in spherical structures (Fig 2D). These spherical structures contained various disorganized organelles, consisted of large amounts of vesicles, damaged mitochondria, and autophagic components (Fig 2D), similar to previously observed megaphagosomes [28]. These structures were also similar to the previously observed LBs [31]. In addition, honeycomb-shaped crystalline arrays as replication particles of CVB3 [32] were observed and were more abundant in α-syn OE dSH-SY5Y cells than in dSH-SY5Y cells (Fig 2E). They were also observed in mouse primary neurons (Fig 2F). Fibrillar structures were observed in CVB3-infected cells. The width and length of these fibrillar structures in dSH-SY5Y cells were approximately 20 nm and 400 nm, respectively (Fig 2G). The fibrils were more numerous and longer in α-syn OE dSH-SY5Y cells than in dSH-SY5Y cells and were not found in α-syn KO dSH-SY5Y cells (Fig 2G). These patterns were also observed in the primary cortical neurons of WT and α-syn transgenic (TG) mice (Fig 2H). These results suggested that CVB3 infection induced the formation of LB-like inclusions in neurons. In addition, damaged mitochondria were analyzed as described previously [33]. In the resting condition, there were no differences in mitochondrial morphology between dSH-SY5Y cells and α-syn OE dSH-SY5Y cells. However, after infection with CVB3, the number of damaged mitochondria was increased in dSH-SY5Y cells and was even more in α-syn OE dSH-SY5Y cells (Fig 2I and 2J), suggesting that mitochondrial damage upon CVB3 infection was accelerated in response to α-syn overexpression.

## α-Syn regulates the maturation of LB-like inclusion bodies induced by CVB3

We analyzed the relationship between LB-like inclusion bodies formed by CVB3 and α-syn in more detail. CVB3 induced the formation of different types of LB-like inclusion bodies over time as evaluated by LC3 staining patterns. We classified them into four stages based on the staining pattern of LC3 (Fig 3A). VP1 of CVB3 was observed, and the stage where intracellular

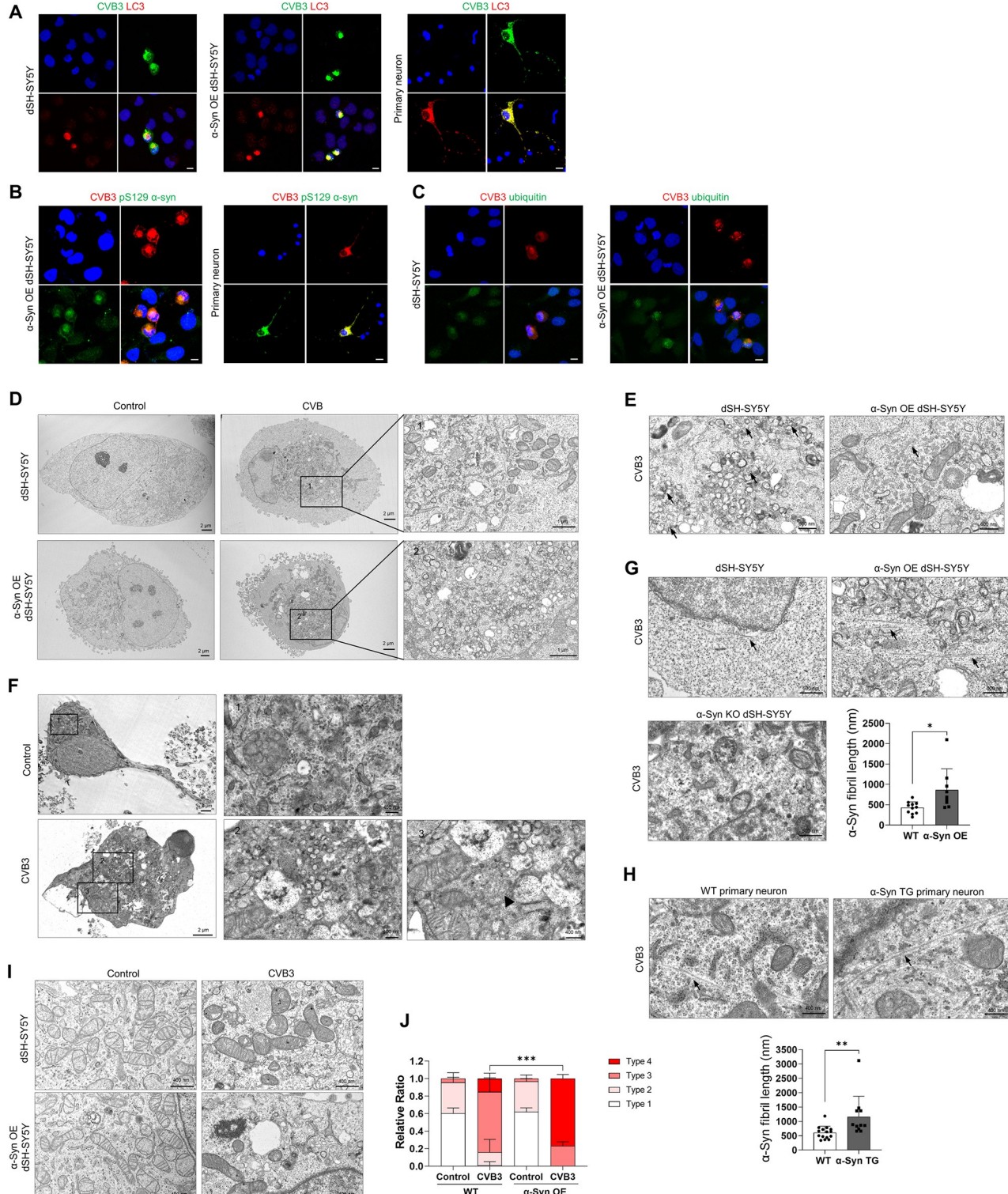

**Fig 2. CVB3 infection induces Lewy body-like inclusion body formation in neurons.** (A) ICC images of WT, α-syn OE dSH-SY5Y cells and mouse primary cortical neurons infected by CVB3 (MOI 0.25 or 5) for 24 h. Cells were immunostained for CVB3 VP1 (green) and MAP1LC3B (LC3) (red). (B) ICC images of α-syn OE dSH-SY5Y cells and mouse primary cortical neurons infected with CVB3 for 24 h. Cells were immunostained for CVB3 VP1 (red) and pSer129 α-syn (green). (C) ICC images of WT and α-syn OE dSH-SY5Y cells infected with CVB3 for 24 h. Colocalization of CVB3 VP1 (red) and ubiquitin (green) was observed. Scale bar indicates 10 μm. Blue indicates Hoechst. (D) Transmission electron microscopy (TEM) images of control and CVB3 infected WT and α-syn OE dSH-SY5Y cells which were infected with CVB3 for 24 h. (E) TEM images of WT and α-syn OE

dSH-SY5Y cells which were infected with CVB3 for 24 h. Arrows indicate viral particles. (F) TEM images of control and mouse primary cortical neurons infected with CVB3 for 24 h. Arrows indicate viral particles. (G) The length of α-syn fibrils between CVB3-infected WT and α-syn OE dSH-SY5Y cells was analyzed. Arrows indicate α-syn fibril-like structures. * P < 0.05, unpaired t-test. (H) The length of α-syn fibrils between CVB3 infected WT and α-syn TG primary neurons was analyzed. Arrows indicate α-syn fibril-like structures. ** P < 0.01, unpaired t-test. (I) TEM images of control and CVB3 infected (24 h) WT and α-syn OE dSH-SY5Y cells. (J) TEM analysis of mitochondrial types of WT and α-syn OE dSH-SY5Y cells infected with CVB3 for 24 h. *** P < 0.001, one-way ANOVA test with Tukey's multiple comparison test.

LC3 morphology does not differ from uninfected cells was defined as stage 1 (Fig 3A1). The stage where the intracellular arrangement of LC3 began to show slight changes was defined as stage 2 (Fig 3A2), and the stage where LC3 began to form a sphere was defined as stage 3 (Fig 3A3). Finally, the stage where LC3 formed a complete sphere with strong intensity was defined as stage 4 (Fig 3A4). These structures also colocalized with pSer129 α-syn as the stage progresses. Over time, the number of inclusion bodies in stage 4 was increased (Fig 3B and 3C), suggesting the maturation of LB-like inclusion bodies. Interestingly, the number of inclusion bodies in stage 4 was higher in α-syn OE SH-SY5Y cells and lower in α-syn KO cells (compared to the control) (Fig 3B–3D). In addition, in α-syn OE SH-SY5Y cells, LC3 was colocalized with inclusion bodies with higher intensity than in the control and α-syn KO cells (Fig 3E and 3F). Inclusion bodies were more condensed and the proportion of CVB3 present in the inclusion bodies was also increased (Fig 3E and 3G–3H), suggesting that α-syn overexpression accelerated the maturation of inclusion bodies whose formation was induced by CVB3.

## α-Syn regulates the replication of CVB3 in neurons

Large autophagosomes induced by CVB3 serve as viral replication complexes [34]. Given that α-syn overexpression accelerated the maturation of inclusion bodies whose formation was induced by CVB3 and that α-syn OE dSH-SY5Y cells displayed greater VP1 intensity in inclusion bodies than dSH-SY5Y cells, we investigated whether α-syn affected the replication of CVB3. CVB3 replication was increased in α-syn OE dSH-SY5Y cells. On the contrary, CVB3 replication was decreased in α-syn KO dSH-SY5Y cells. Primary neurons from α-syn TG mice also showed similar results (Fig 4A), consistent with the TEM analysis (Fig 2E). Infection with the CVB3 variant co-expressing EGFP (CVB3-EGFP) produced increased number of EGFP-positive cells and enhanced the EGFP intensity in α-syn OE SH-SY5Y cells (Fig 4B–4E). Additionally, when SH-SY5Y cells overexpressing mCherry only or α-syn-mCherry were infected with CVB3-EGFP, the intensity of EGFP was positively proportional to that of mCherry in α-syn-mCherry OE SH-SY5Y cells, but not in mCherry only OE SH-SY5Y cells (Fig 4F and 4G). Additionally, when both cells were infected with CVB3 for 30 h, the cytotoxicity of CVB3 infection was more severe in α-syn OE SH-SY5Y cells (Fig 4H), suggesting that α-syn promotes CVB3 replication and related cytotoxicity.

## CVB3 and α-syn differentially regulate autophagic activity

CVB3 inhibits the fusion of autophagosomes with the lysosomes and uses autophagosomes as replication complexes [34]. To confirm this, we monitored autophagic activity [35, 36]. LC3II levels were increased upon CVB3 infection. However, it was not increased further by treatment with bafilomycin A1 (BafA1) (Fig 5A), suggesting that CVB3 inhibited the late stage of the autophagic process. This was further supported by a significant reduction in lysosomes, as evaluated by LysoTracker staining, in CVB3-infected dSH-SY5Y cells (Fig 5B). Further, LC3II levels were increased and treatment with BafA1 further increased LC3II levels in α-syn OE dSH-SY5Y cells (Fig 5C). The levels of p62 were decreased (Fig 5D), suggesting that α-syn overexpression resulted in increased autophagic flux, which agreed with the findings of a

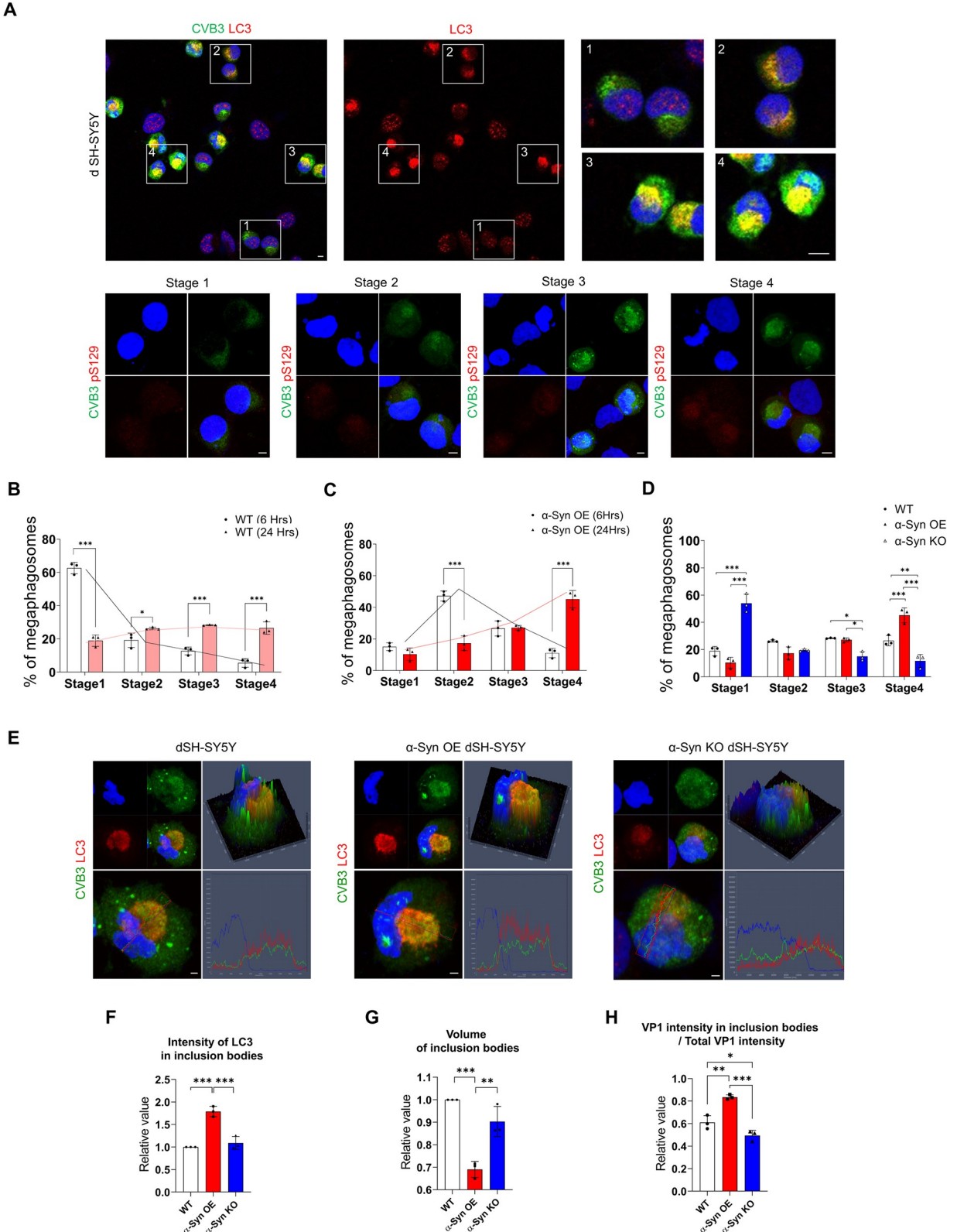

**Fig 3. α-Syn regulates the maturation of Lewy body-like inclusion bodies induced by CVB3.** (A) ICC images of dSH-SY5Y cells infected with CVB3 (MOI 0.25) for 24 h. Cells were immunostained for CVB3 VP1 (green) and MAP1LC3B (LC3) (red). Higher magnification of images enclosed in numbered white boxes indicates stages of inclusion bodies based on the staining pattern of LC3. Cells were immunostained for

CVB3 VP1 (green) and pSer129 α-syn (red). Scale bar indicates 10 μm. Blue indicates Hoechst. (B) ICC image analysis of the ratio of inclusion bodies in each stage in dSH-SY5Y cells infected with CVB3 for 6 and 24 h. *** $P < 0.001$, * $P < 0.05$, two-way ANOVA test with Sidak's multiple comparison test. (C) ICC image analysis of the ratio of inclusion bodies in each stage of α-syn OE dSH-SY5Y cells infected with CVB3 for 6 and 24 h. *** $P < 0.001$, two-way ANOVA test with Sidak's multiple comparison test. (D) ICC image analysis of the percentage of inclusion bodies in each stage in WT, α-syn-OE, and α-syn KO dSH-SY5Y cells which were infected with 0.25 MOI of CVB3 for 24 h. *** $P < 0.001$, ** $P < 0.01$, * $P < 0.05$, two-way ANOVA test with Sidak's multiple comparison test. (E) ICC images of VP1 (green) and LC3 (red) colocalization patterns in WT, α-syn OE and α-syn KO dSH-SY5Y cells infected with CVB3 for 24 h. Images are representative of independent experiments (n = 3). Scale bar indicates 2 μm. Blue indicates Hoechst. (F and G) ICC image analysis of LC3 intensity and volume of inclusion bodies in each stage in WT, α-syn-OE and α-syn KO dSH-SY5Y cells infected with CVB3 for 24 h. Values are derived from three independent experiments (n = 3). *** $P < 0.001$, ** $P < 0.01$, one-way ANOVA test with Tukey's multiple comparison test. (H) ICC image intensity analysis of VP1 in inclusion bodies per total VP1 intensity in each stage in WT, α-syn-OE and α-syn KO dSH-SY5Y cells that were infected with CVB3 for 24 h. Values are derived from three independent experiments (n = 3). *** $P < 0.001$, ** $P < 0.01$, * $P < 0.05$, one-way ANOVA test with Tukey's multiple comparison test.

previous study [37]. We further analyzed open source database information. Overexpression of human α-syn using a lentiviral vector in mouse midbrain neurons revealed that the cluster of transcriptome was characterized by a "positive regulation of macroautophagy" (GO:0016239) in gene ontology (GO) analysis compared to control (GSE70368) (S3 Fig). In addition, "autophagosome maturation" (GO:0097352) in induced pluripotent stem cells (iPSCs) of α-syn (*SNCA*) triplicated family (GSE30792) and "lysosome organization" (GO:0007040) in the mouse striatum tissue of human α-syn TG mice (GSE116010) were also higher than in controls (S3 Fig). These findings supported our data. Compared with the control, CVB3 infection further increased LC3 II levels, and treatment with BafA1 induced similar results in α-syn OE dSH-SY5Y cells (Fig 5E and 5F). These results suggested that α-syn overexpression promoted autophagic flux and accelerated the formation of autophagosomes, which provided more replication centers for CVB3.

## α-Syn regulates CVB3 replication in mice brains

To confirm whether CVB3 also interacts with α-syn in the brain, CVB3 was intraperitoneally injected into mice. α-Syn expression in TG mice was driven by the NSE promoter and was 1.5 times higher than that in WT mice (S4A Fig). In our experimental condition, no clinical or histological abnormalities were observed in TG mice without CVB3 infection. At day 7 postinfection (PI), the detection of CVB3 was more pronounced in the brains of α-syn TG mice, compared to that in WT mice (Fig 6A). The *in vivo* data were consistent with the *in vitro* data. Immunohistochemistry (IHC) of the brains from WT mice revealed the presence of CVB3 in the olfactory area, anterior cingulate area, lateral septal nucleus, hippocampal region, fimbria, corticospinal tract, and hypothalamus, mainly along the ventricles, which was colocalized with the microglial marker, Iba-1. In addition, CVB3 was detected in the hippocampus, lateral thalamus, and midbrain, and colocalized with the neuronal marker, Tuj-1. CVB3 did not colocalize with the astrocyte marker, GFAP (Fig 6B and 6C), suggesting that CVB3 was detected in the brain and infected the microglia and neurons, but not astrocytes.

In addition, CVB3 was observed in the neurons located in the hippocampus, lateral thalamus, and midbrain at day 4 PI. At day 7 PI, the number of neurons infected with CVB3 was increased compared to that in mice at day 4 PI (Figs 6D and S4B). At day 28 PI, the number of midbrain neurons infected with CVB3 was clearly increased whereas the number of neurons located in other regions infected by CVB3 was relatively decreased (Fig 6D). The pattern of infected neurons was more accelerated in α-syn TG mice than that in WT mice. At day 4 PI, the number of neurons infected with CVB3 was not significantly different between WT mice and α-syn TG mice, but relatively more neurons infected with CVB3 were observed in the hippocampus of α-syn TG mice, compared to that of WT mice (Fig 6D). At day 7 PI, the pattern of infected neurons in the TG mice was comparable with that in WT mice at day 28 PI

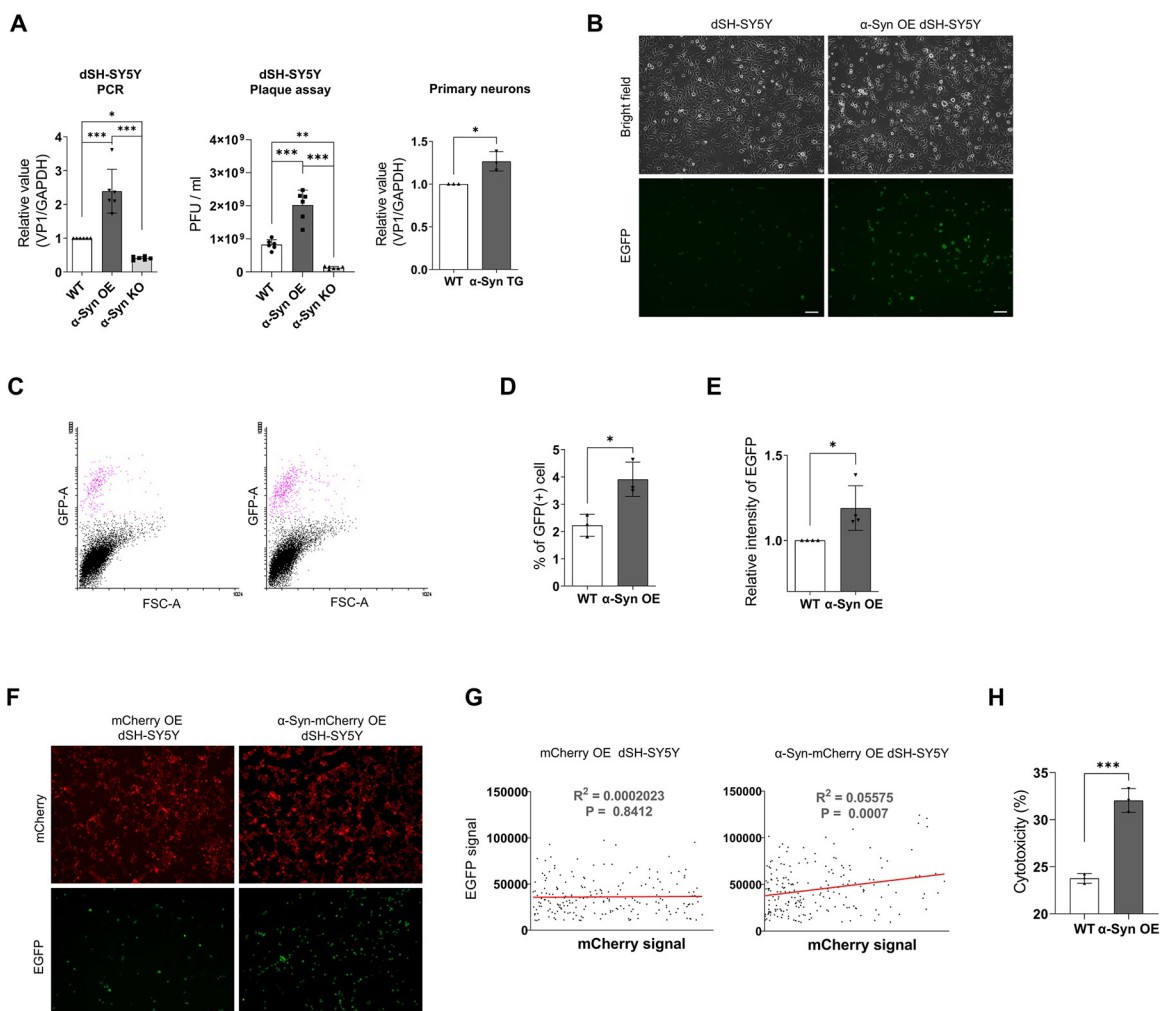

**Fig 4. α-Syn regulates replication of CVB3 in neurons.** (A) The relative levels of VP1 as analyzed by PCR and plaque assay between WT, α-syn OE, and α-syn KO dSH-SY5Y cells that were infected with CVB3 (MOI 0.25) for 24 h. In case of plaque assay, each cell was infected with CVB3 (MOI 1) for 30 h. The relative levels of VP1—as analyzed by PCR—between WT and α-syn TG mice primary cortical neurons that were infected with CVB3 (MOI 5) for 24 h. Values are derived from six or three independent experiments (n = 6 or 3). *** P < 0.001, ** P < 0.01, * P < 0.05, one-way ANOVA test with Tukey's multiple comparison test and unpaired t-test. (B) Fluorescence microscopy images of WT and α-syn OE dSH-SY5Y cells that were infected with 0.6 MOI CVB3 variant co-expressing EGFP (CVB3-EGFP) for 24 h. Scale bar indicates 50 μm. (C, D and E) Flow cytometric analysis. The number of EGFP-positive cells (D) and intensity (E) were analyzed using WT and α-syn OE dSH-SY5Y cells that were infected with CVB3-EGFP (MOI 0.6) for 24 h. Values are derived from three or four independent experiments (n = 3 or 4). * P < 0.05, unpaired t-test. (F) Fluorescence microscopy images of mCherry and α-syn-mCherry OE dSH-SY5Y cells infected with CVB3-EGFP (MOI 0.6) for 24 h. Values and values are derived from three independent experiments (n = 3). Scale bar indicates 50 μm. (G) Linear regression analysis of mCherry and EGFP intensity of mCherry and α-syn-mCherry OE dSH-SY5Y cells infected with CVB3-EGFP (MOI 0.6) for 24 h. Values are representative of three independent experiments. (H) Cytotoxicity analysis of WT and α-syn OE dSH-SY5Y cells infected with CVB3 (MOI 0.25) for 30 h by lactate dehydrogenase assay. Values are derived from three independent experiments (n = 3). *** P < 0.001, unpaired t-test.

(Figs 6D and S4B). The number of CVB3-infected microglia increased at day 7 PI (Figs 6E and S4C). At day 28 PI, the number of CVB3-infected microglia was similar to that at day 7 PI (Figs 6E and S4C). The number of CVB3-infected microglia was also higher in TG mice than in WT mice during the same period (Figs 6E and S4C). In addition, activation of microglia was observed upon CVB3 infection (S5 Fig).

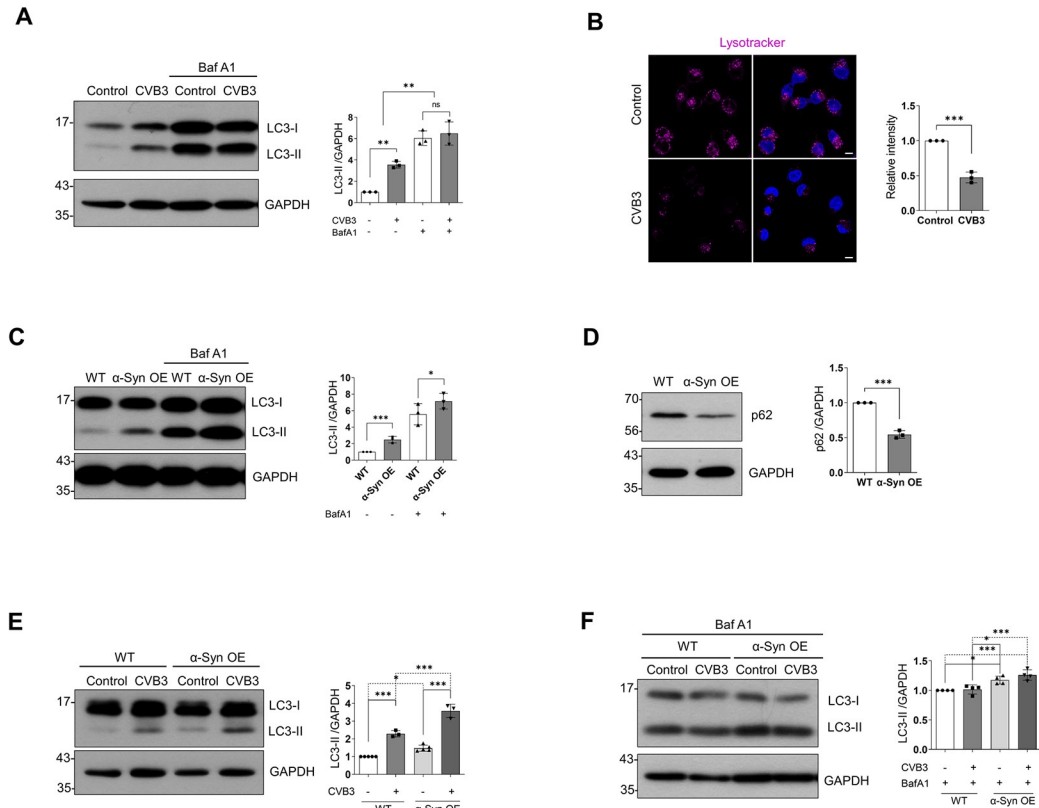

**Fig 5. CVB3 and α-syn differentially regulate autophagic activity.** (A) Samples of control and CVB3 infected (0.25 MOI for 24 h) dSH-SY5Y cells treated with dimethylsulfoxide (DMSO) or 50 nM bafilomycin A1 (BafA1) were lysed and western blotting was performed using LC3 antibody. Protein levels were quantified by densitometry. Values are derived from three independent experiments (n = 3). ** P < 0.01, one-way ANOVA test with Tukey's multiple comparison test. (B) ICC images of dSH-SY5Y cells infected with CVB3 (MOI 0.25) for 24 h. Fluorescence was seen using LysoTracker. Values are derived from three independent experiments (n = 3). Scale bar indicates 10 μm. *** P < 0.001, unpaired t-test. (C) The relative expression level of p62 between WT and α-syn OE dSH-SY5Y cells. Protein levels were quantified by densitometry. Values are derived from three independent experiments (n = 3). *** P < 0.001, unpaired t-test. (D) The relative expression level of LC3 between WT and α-syn OE dSH-SY5Y cells treated with DMSO or 50 nM BafA1 for 24 h. Protein levels were quantified by densitometry. Values are derived from three independent experiments (n = 3). *** P < 0.001, * P < 0.05, one-way ANOVA test with Tukey's multiple comparison test. (E and F) The relative expression levels of LC3 between WT (control and CVB3) and α-syn OE (control and CVB3) dSH-SY5Y cells with DMSO or 50 nM BafA1. Cells were infected with 0.25 MOI CVB3 for 24 h. Protein levels were quantified by densitometry. Values are derived from three independent experiments (n = 3). *** P < 0.001, * P < 0.05, one-way ANOVA test with Tukey's multiple comparison test.

At day 28 PI, the expression of cleaved caspase 3, a marker for apoptosis, was focally observed throughout the brain of WT mice (Figs 6F and S6). This observation was also evident in TG mice at day 7 PI, which was comparable to that in WT mice at day 28 PI (Fig 6G). Additionally, dopaminergic neurons located in the substantia nigra were infected with CVB3 (Fig 6H). The number of tyrosine hydroxylase (TH)-positive cells was slightly decreased in the substantia nigra of CVB3-infected mice at day 28 PI, compared with the control, which was also comparable to that in TG mice at day 7 PI (Fig 6I and 6J).

We then monitored the survival rate of both WT and α-syn TG mice. In the absence of CVB3 infection, there was no difference in weight gain between WT and α-syn TG mice, and both groups of mice survived in our experimental condition. However, after infection with CVB3, the survival of α-syn TG mice was poorer than that of WT mice. Weight loss was more severe in α-syn TG mice. These findings suggested that α-syn TG mice were more susceptible

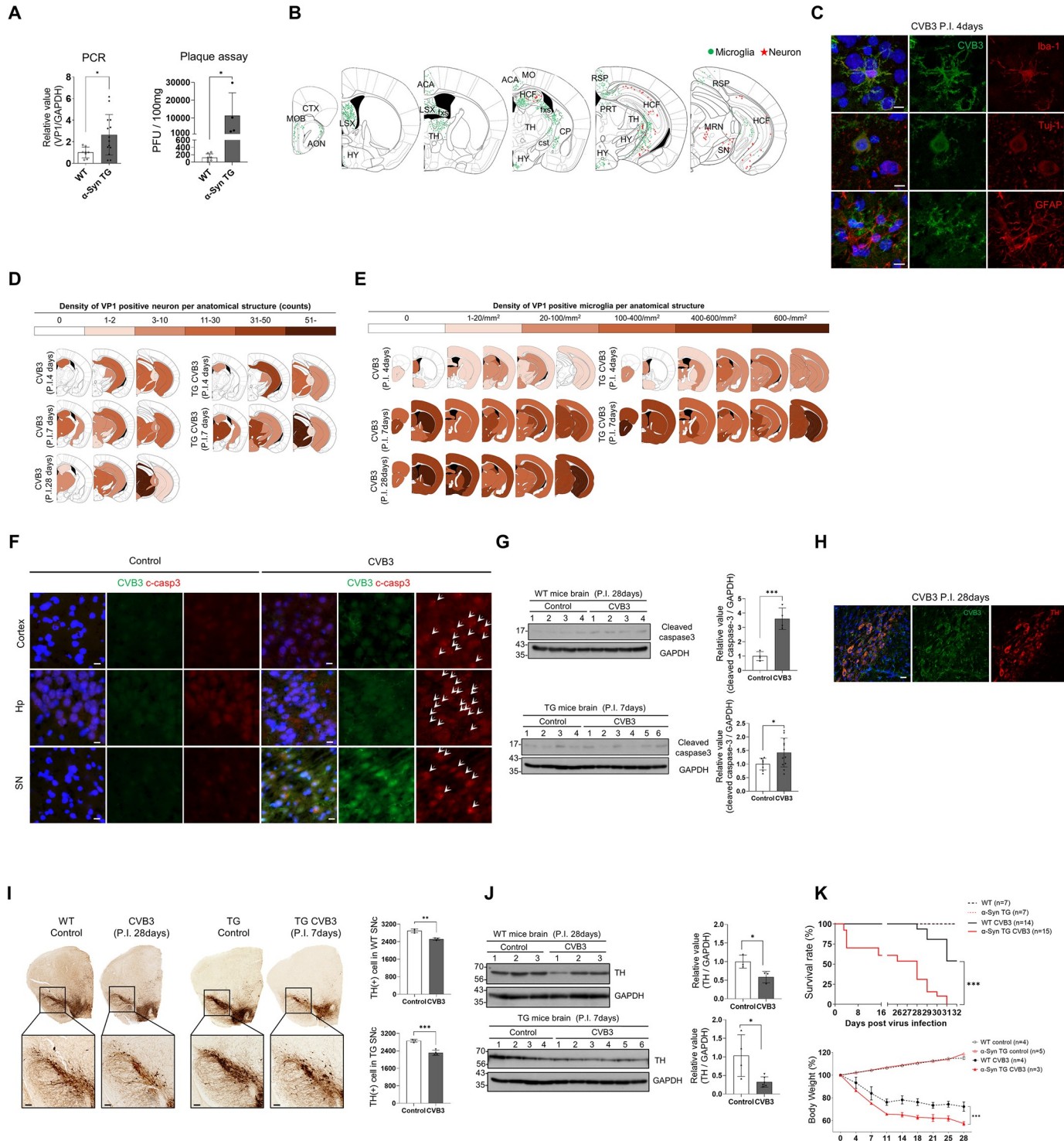

**Fig 6. α-Syn regulates CVB3 replication in mice brains.** (A) The relative levels of VP1—as analyzed by PCR—between WT (n = 8) and α-syn TG (n = 14) mice brains, and plaque assay between WT (n = 7) and α-syn TG (n = 4) mice brains at day 7 post infection (PI). Mice were infected by IP injection of $1.0 \times 10^6$ PFUs of CVB3. * P < 0.05, unpaired t-test. (B) Immunohistochemistry (IHC) images showing the localization of VP1-positive cells in mouse brain regions at day 4 PI. Microglia and neurons are indicated as green and red asterisks, respectively. (Abbreviations: mouse brain regions: CTX-cortex, MOB-main olfactory bulb, AON-anterior olfactory nucleus, LSX-lateral septal complex, HY-hypothalamus, ACA-anterior cingulate area, fxs-fornix system, MO-somatomotor area, HCF-hippocampal formation, TH-thalamus, CP-caudoputamen, Cst-corticospinal tract, RSP-retrosplenial area, PRT-pretectal region, MRN-midbrain reticular nucleus, SN-substantia nigra). (C) IHC images of CVB3 infected mice brains at day 4 PI (IP injection of $1.0 \times 10^6$ PFUs of CVB3). Colocalization of CVB3 VP1 (green) with

Iba-1 (red), Tuj-1 (red), and GFAP (red) were observed. Scale bar indicates 10 μm. Blue indicates Hoechst. (D and E) Time series (at day 4,7 and 28 PI) anatomic diagram illustrating the pattern of CVB3 infection (IP injection of $1.0 \times 10^6$ PFUs of CVB3). Tuj-1 (D) and Iba-1 (E) positive cell densities, which were colocalized with VP1 in CVB3 infected WT and α-syn TG mice are expressed in the corresponding color of the indicated number based on the anatomical structure. (F) IHC images of control and CVB3 infected mice brains at day 28 PI (IP injection of $1.0 \times 10^6$ PFUs of CVB3). Arrows indicate cleaved caspase-3 (c-caspase-3) positive cells. (Abbreviations of mouse brain structure: Hp: hippocampus, SN: substantia nigra). Scale bar indicates 10 μm. Blue indicates Hoechst. (G) Western blotting was performed using control and CVB3 infected (IP injection of $1.0 \times 10^6$ PFUs of CVB3) WT (at day 28 PI) and α-syn TG (at day 7 PI) mice brain samples. Protein levels were quantified by densitometry. *** $P < 0.001$, * $P < 0.05$, unpaired t-test. (H) IHC images of CVB3 infected mice brains at day 28 PI (IP injection of $1.0 \times 10^6$ PFUs of CVB3). Colocalization of CVB3 VP1 (green) with tyrosine hydroxylase (TH) (red) was observed. Scale bar indicates 20 μm. Blue indicates Hoechst. (I) TH-positive diaminobenzidine (DAB) image of control and CVB3 infected mice brain at day 28 PI (WT) and at day 7 PI (α-syn TG). The number of TH-positive cells in the substantia nigra pars compacta (SNpc) between control and CVB-infected mice were analyzed. Scale bar indicates 100 μm. *** $P < 0.001$, ** $P < 0.01$, unpaired t-test. (J) Western blotting was performed using control and CVB3-infected (IP injection of $1.0 \times 10^6$ PFUs of CVB3) WT (at day 28 PI) and α-syn TG (at day 7 PI) mice. Protein levels were quantified by densitometry. * $P < 0.05$, unpaired t-test. (K) Kaplan-Meier survival curves and body weight loss curve of (IP injection of $1.0 \times 10^6$ PFUs of CVB3) control, CVB-infected WT and α-syn TG mice. *** $P < 0.001$, Log-rank (Mantel-Cox) test (survival curve analysis). *** $P < 0.001$, two-way ANOVA test (weight loss curve analysis). Figure 6B, 6D, and 6E were modified after downloading the open source brain image (http://labs.gaidi.ca/mouse-brain-atlas).

to CVB3 infection (Fig 6K). CVB3 is a cardiotropic virus that induces myocarditis [16]. Therefore, we monitored myocardial damage and replication of CVB3 in the heart. Heart damage and the levels of VP1 in the myocardium of both mice groups were similar (S7A and S7B Fig). In addition, CVB3 cause extensive pancreatic tissue damage in experimental animal model [38, 39]. Histological examinations of the pancreas using hematoxylin and eosin staining indicated pancreatic tissue damage with inflammation. However, no difference between both mice groups was observed (S7C Fig). Endogenous α-syn expression in the heart and the pancreas was much lower than that in the brain and there was no difference of endogenous α-syn expression in the heart and the pancreas from both WT and TG mice, because α-syn overexpression was controlled by NSE promoter (S7D Fig). These findings suggested that the difference in the survival rate of both mice groups may not be due to cardiac damage or pancreatic damage. Histological examinations of the heart, liver, and spleen also indicated no difference between WT and α-syn TG mice (S7C Fig). These results suggested that CVB3 infection in the mouse brain caused neuronal death, including loss of dopaminergic neurons located in the substantia nigra, and that α-syn accelerated the replication of CVB3 and CVB3-induced neuronal death.

## CVB3 induces the formation of α-syn inclusions in mice brains

Next, we examined the colocalization of α-syn and CVB3 VP1 in the brains of mice infected with CVB3. We did not observe the colocalization of α-syn and CVB3 VP1 in the brains of WT or α-syn TG mice. Instead, at day 7 PI, a few neurons containing α-syn accumulation in the cell body were observed in the interpeduncular nucleus of WT mice upon CVB3 infection and the intensity of α-syn was lower than that of the control (Fig 7A). However, α-syn accumulations in the cell body did not colocalize with pSer129 α-syn. In α-syn TG mice, more neurons containing α-syn accumulation in the cell body were observed in the same region (Fig 7A), although they also did not colocalize with pSer129 α-syn. We performed western blotting to detect pSer129 α-syn. At day 28 PI in WT mice, pSer 129 α-syn was detected at low levels and a decrease in α-syn expression was observed (Fig 7B). Similar results were observed in α-syn TG mice at day 7 PI, suggesting that α-syn accumulation in the cell body was induced upon CVB3 infection and was accelerated in response to α-syn overexpression (Fig 7C).

## Discussion

In spite of extensive research, the mechanism underlying PD pathogenesis remains elusive. Both genetic and environmental factors and their crosstalk are suspected to contribute to the pathogenesis of PD [40]. LBs are the main pathological hallmarks of PD. Additionally, Lewy

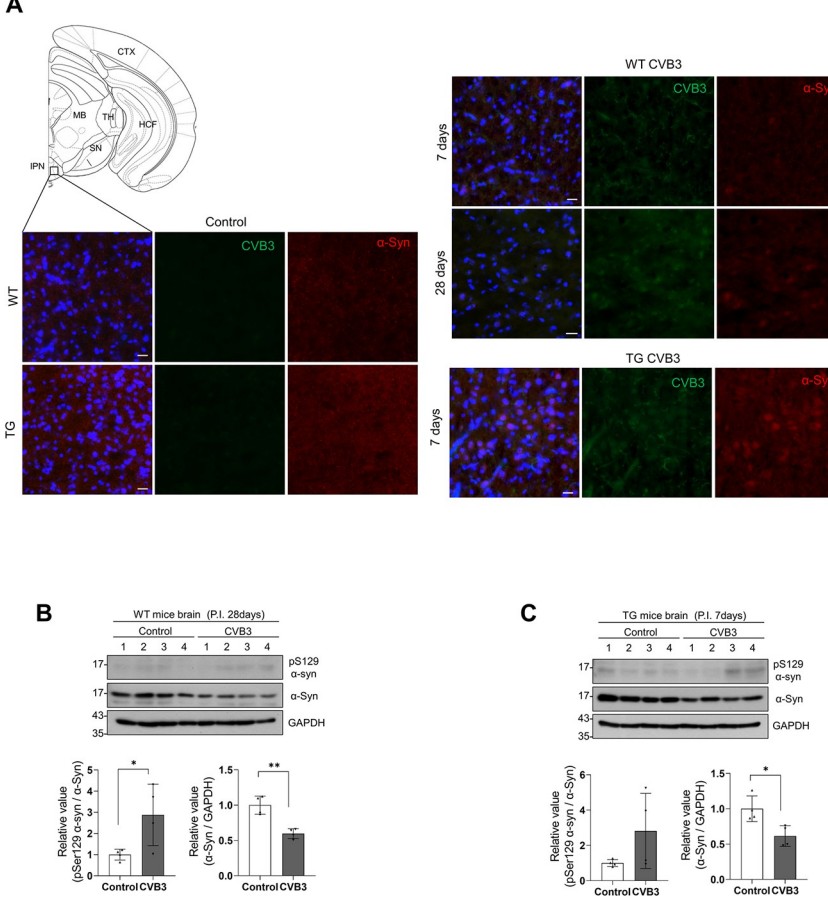

**Fig 7. CVB3 induces α-syn inclusions in mice brains.** (A) IHC images of control, CVB3 infected WT and α-syn TG mice brains at indicated days PI (IP injection of $1.0 \times 10^6$ PFUs of CVB3). Scale bar indicates 20 μm. Blue indicates Hoechst. (Abbreviations of mouse brain regions: CTX-cortex, HCF-hippocampal formation, MB-midbrain, TH-thalamus, SN-substantia nigra, IPN-interpeduncular nucleus). (B and C) Western blotting was performed using control, CVB3 (IP injection of $1.0 \times 10^6$ PFUs of CVB3) infected WT and α-syn TG mice brain samples at indicated days PI. Protein levels were quantified by densitometry. *** $P < 0.001$, ** $P < 0.01$, * $P < 0.05$, unpaired t-test. Fig 7A was modified after downloading the open source brain image (http://labs.gaidi.ca/mouse-brain-atlas).

pathology progressively involves more regions of the nervous system as the disease advances, and it manifests prior to the appearance of motor symptoms in PD [8]. Accordingly, it is important to identify the factors that initiate Lewy pathology to understand the pathogenesis of PD. Several factors are suspected to trigger Lewy pathology. Several environmental factors that contribute include pathogens such as influenza virus, environmental pollutants like pesticides, heavy metals, and head trauma [41]. In particular, viral infection has long been considered as a risk factor for neurodegenerative diseases [13]. The present data demonstrate that CVB3 infection and α-syn expression have a mutual effect on each other.

With respect to the influence of CVB3 on α-syn, CVB3 infection induces very large autophagy-related structures ranging from 10–20 μm in diameter in neurons. These structures colocalize with LC3 and have a similar morphology as that of megaphagosomes observed in pancreatic acinar cells [28]. They also colocalized with α-syn, pSer129 α-syn, and ubiquitin, suggesting a resemblance to LBs. LBs exhibit significant morphological diversity and are heterogeneous in their shape, biochemical composition, and organization [30]. Examination of the brains of patients with PD using super-high resolution microscopy based on stimulated

emission depletion (STED) revealed LBs with crowded organelles and lipid membranes. This prompted the proposal that α-syn may modulate the compartmentalization and function of membranes and organelles in LB-affected cells [31]. A recent report demonstrated the formation of filament-like structures accompanied by the sequestration of lipids, organelles, and endomembrane structures using a seeding-based model of α-syn fibrillization, which recapitulated the features of LBs observed in the brains of patients with PD [42]. Likewise, our TEM findings also suggested that CVB3 infection resulted in the clustering of several organelles in the perinuclear space in neurons. The crowded organelles contained damaged mitochondria and many fibrillar structures surrounding the organelles. The number of fibrillar structures in α-syn OE dSH-SY5Y cells was higher than that in dSH-SY5Y cells. These structures in α-syn OE dSH-SY5Y cells were also longer than those in dSH-SY5Y cells. These structures were barely detected in α-syn KO cells. The α-syn fibrils exhibit 20 nm in diameter *in vitro* [43]. These observations suggested that these fibrillar structures may be α-syn fibrils, although we could not confirm this by immune-EM analysis. Additionally, LB formation involves several stages [44, 45]. CVB3 induced different types of LB-like inclusion bodies over time, which may reflect the maturation of the inclusion bodies. In α-syn OE cells, the maturation of these inclusion bodies was accelerated and they were more condensed, suggesting that α-syn may regulate the maturation of inclusion bodies as a major component. In addition, mitochondrial damage was induced by CVB3, which was more in α-syn OE dSH-SY5Y cells. The observations are supported in part by previous studies demonstrating that α-syn localizes to the mitochondria and α-syn OE cells exhibit mitochondrial dysfunction [46–48]. We demonstrated that CVB3 inhibited the late stage of autophagy in dSH-SY5Y cells, consistent with the findings of a previous study [28]. It is well known that α-syn is a substrate for autophagic degradation [49–51], although other mechanisms have also been reported to be involved in α-syn degradation [52]. Dysfunctional late stage autophagy has been reported to be associated with α-syn accumulation [53]. Accordingly, CVB3 might employ the autophagy machinery to induce the formation of LB-like inclusions associated with α-syn.

Although CVB3 induced the formation of large inclusion bodies containing α-syn, its expression was decreased. We confirmed this using *in vitro* and *in vivo* model systems and open source data from CVB3-infected hearts of mice. In particular, neighboring cells may be more affected. It might not be due to cytotoxicity or the host shutoff phenomenon upon CVB3 infection. Nevertheless, there effects cannot be disregarded completely. The differential regulation of α-syn expression requires further investigation. A previous report demonstrated that WNV induced α-syn expression, and α-syn was proposed as a viral restriction factor [21]. We also observed that treatment with polyIC increased α-syn expression in dSH-SY5Y cells, suggesting that α-syn expression can be regulated by viral infection. However, the decrease in α-syn expression may be CVB3-specific. Interestingly, decreased α-syn mRNA in the brains of patients with PD has been described [54, 55]. These findings are contentious owing to several technical issues, including sampling and normalization methods. Our analysis of open data sources from patients with PD also confirmed it. Therefore, CVB3 infection in patients may indicate the onset of PD.

When we infected mice with CVB3, infection first appeared in the region of several anatomical structures along the ventricles, suggesting the peripheral route of CVB3 infection into the CNS. Neuron and microglia infection spread to other regions with time. In neurons, CVB3 was observed in the hippocampus, lateral thalamus, and midbrain in the brains of mice at day 4 PI. Although we observed the colocalization of CVB3 with α-syn *in vitro*, we could not observe the colocalization in the brains of mice infected with CVB3. Instead, we observed α-syn accumulation in the cell bodies of neurons located in the midbrain of WT mice at day 28 PI, and more neurons containing cytosolic α-syn accumulation were observed in α-syn TG

mice at day 7 PI. In addition, western blot analysis indicated that the expression of pSer129 α-syn was slightly increased in brains infected with CVB3. In a previous report [41], the authors proposed that triggers alone are usually insufficient for the development of PD. Triggers often act transiently, with the triggering event lasting a few weeks or months and occurring relatively early in the life of individuals who develop PD. Accordingly, CVB3 infection may act transiently. It alone may not induce LB formation in the brain, unlike *in vitro* observations. Alternatively, neuronal α-syn aggregation observed in our *in vivo* system may not reach the same maturation stage as the PD LBs, since our *in vivo* model system is not adequate for prolonged observations due to the high mortality rate of C57BL/6 mice in response to CVB3 infection [56]. The long-term consequences of CVB3 infection in the CNS are largely unknown. However, these viruses persist at extremely low levels in the adult CNS [57]. Nevertheless, the presence of viral RNA by itself is potentially pathogenic in some cases such as schizophrenia [58] and amyotrophic lateral sclerosis [59]. The very low viral titer in the CNS might not be sufficient to observe the distribution of CVB3 infection and the colocalization between CVB3 and α-syn in the brains of mice infected with CVB3 using IHC. Interestingly, CVB3 infected BALB/c mice, which are more susceptible to chronic CVB3 infection, reportedly showed TDP-43 aggregation in the hippocampal region at 90 days PI [60]. In addition, it has been reported that cytosolic aggregates as well as soluble oligomers, which were not observed in healthy controls, were observed in the heart of patients with dilated cardiomyopathy, which is suspected to be caused by CVB3 infections [61, 62]. Congo red merged islet amyloid polypeptide was also seen in pancreatic biopsies of patients with type 1 diabetes, suggesting that the formation of these aggregates may be induced by enteroviruses [63]. Accordingly, virus-induced formation of intracellular protein inclusions is not restricted to neurons; rather, this is a general phenomenon.

In this study, we observed that α-syn regulated CVB3 replication. Overexpression of α-syn resulted in increased CVB3 replication and CVB3-induced cytotoxicity. We confirmed this *in vivo* in the brains of α-syn TG mice. CVB3 replication was increased and the distribution of CVB3-infected regions was also expanded in α-syn TG mice compared to that in control mice. In addition, CVB3 infection induced neuronal cell death including loss of dopaminergic neurons in the substantia nigra. Interestingly—supporting our observation—patients with chronic EV71 encephalitis whose symptoms persisted for more than 2 months displayed damage in most of the midbrain, including the substantia nigra [64]. Dopaminergic neuronal cell death in the substantia nigra was also accelerated in α-syn TG mice. Furthermore, the survival rate of α-syn TG mice was less compared to that of WT mice. A previous report demonstrated that α-syn expression inhibits WNV growth and replication, resulting in increased mortality of α-syn knock-out mice [21], which contradicts our findings, which demonstrate that α-syn TG mice show enhanced CVB3 replication and a lower survival rate after CVB3 infection. We cannot completely explain the discrepancy between these results. The residual amount of α-syn in the brain may be important for the regulation of viral infection. Nevertheless, this could be due to the distinct autophagy-related pathways engaged by the virus. Previous studies have suggested that CVB3 uses autophagosomes as its replication centers by inhibiting the fusion of autophagosomes with the lysosomes [28, 65, 66]. The induction of autophagy in neurons was also associated with increased CVB3 replication [65–67]. An increase in α-syn expression accelerated the autophagic flux. It has also been reported that α-syn overexpression is sufficient to impair autophagosome maturation in a Drosophila model system [68]. This environment favors the replication of CVB3. In contrast, autophagy is known to inhibit WNV replication [69, 70]. Accordingly, it may be virus-specific, and autophagy in both viruses may explain this discrepancy.

The epidemiological link suggests that viral exposure over time may increase the risk of PD, although it is unclear whether any specific viral infection causes PD. It could be related to direct virus-induced cytotoxicity or virus-related inflammation [13, 71, 72]. Influenza viral infections induce parkinsonian symptoms and a significant increase in phosphorylation and aggregation of α-syn [73, 74]. Repeated viral infection may induce α-syn expression or/and α-syn aggregation, and chronic viral infection induces further inflammation, which may initiate and lead to the progression of PD. Likewise, we observed that α-syn responded to CVB3 infection. CVB3 infection regulated α-syn expression and aggregation. α-Syn may function as a defense mechanism against viral infection. The finding that CVB3 infection is associated with α-syn suggests an unexpected role of α-syn in the pathogenesis of PD. Further studies are needed to explore this in more detail.

In conclusion, we investigated the relationship between CVB3 infection and α-syn. CVB3 infection induced α-syn-associated inclusion body formation in neurons which might act as a trigger for PD. These inclusion bodies contained clustered organelles including damaged mitochondria with α-syn fibrils. α-Syn overexpression accelerated inclusion body formation and induced more concentric inclusion bodies. Brains of CVB3 infected mice harbored α-syn aggregates in the cell body of the midbrain. The data indicate that CVB3 infection blocks the late stage of autophagy, and induces inclusion body formation containing α-syn fibrils. Overexpression of α-syn favors CVB3 replication and related cytotoxicity. The survival rate of α-syn TG mice was poor. CVB3 replication was more extensive in these mice, which further induced neuronal cell death, including loss of dopaminergic neurons. α-Syn overexpression accelerated autophagic flux, which favored the replication of CVB3. Taken together, our findings clarify the mechanism of LB formation and the pathogenesis of PD associated with CVB3 infection.

## Materials and methods

### Ethics statement

All procedures were conducted according to the guidelines established by the Ajou University School of Medicine Ethics Review Committee (IACUC No. 2016–0047).

### Antibodies and reagents

Antibodies against α-syn were purchased from Abcam (#ab138501, Cambridge, UK), BD Biosciences (#610786, Franklin Lakes, NJ), and Genetex (#GTX112799, Santa Barbara, CA). Antibodies against pSer129 α-syn (#ab51253), Tuj-1 (#ab18207) and Enterovirus 71 (#ab36367) were obtained from Abcam. Antibody against VP1 of CVB3 was purchased from Millipore (#MAB948, Danvers, MA). Antibody against LC3 was purchased from Sigma-Aldrich (#L8918, St. Louis, MO). Antibody against p62 was purchased from BD Biosciences (#610832). Antibody against cleaved caspase-3 was purchased from Cell Signaling Technology (#9664, Beverly, MA). Antibody against Iba-1 was purchased from Wako (#019–19741, Richmond, VA). Antibody against glial fibrillary acidic protein (GFAP) was purchased from Neuromics (#RA22101, Montreal, QC). Antibody against glyceraldehyde 3-phosphate dehydrogenase (GAPDH) was purchased from Santa Cruz Biotechnology (#SC-32233, Santa Cruz, CA). Lyso-Tracker (#L12492) and Lipofectamine2000 (#11668–019) were obtained from Invitrogen (Carlsbad, CA). Retinoic acid (RA, #R2625), bafilomycin A1 (Baf A1, #B1793), polyinosine: polycytidylic acid (poly IC, #P0913), and Evans Blue Dye (EBD, #E2129) were purchased from Sigma-Aldrich. Sudan Black B was purchased from Tokyo Chemical Industry (#4197–2505, Tokyo, Japan).

## Animals

α-Syn transgenic (TG) mice overexpressing human α-syn under the control of the neuron specific enolase (NSE) promoter (C57BL/6N-Tg (NSE-h a Syn) Korl) were donated by the National Institute of Food and Drug Safety Evaluation (NIFDS, Cheongju, Korea). Wild-type (WT) littermates or WT C57BL/6N mice (DBL, Eumseong, Korea) were used as controls.

## Cell culture

α-Syn, mCherry and α-syn-mCherry overexpressing SH-SY5Y cells were generated as described previously [75]. α-Syn KO SH-SY5Y cells were generated using the lentiCRISPR system [76] (human α-syn annealed oligonucleotides; Oligo 1: 5′- TGTAGGCTCCAAAAC-CA AGG-3′, Oligo 2: 5′- CCTTGGTTTTGGAGCCTACA -3′). Oligonucleotides were designed using an online gRNA design tool (https://chopchop.cbu.uib.no). Individual clones were diluted from the transfected population, isolated, and selected using genomic sequencing and western blotting. The cells were grown in Dulbecco's modified Eagle medium (DMEM) supplemented with 10% fetal bovine serum (FBS) at 37˚C in a humidified atmosphere containing 5% $CO_2$ and 95% air. Primary cortical neurons were isolated from one-day-old pups of WT and α-syn TG mice and cultured in neurobasal medium (#21103–049, Invitrogen) with Gluta-MAX-I (#35050061, Thermo Fisher Scientific, Waltham, MA), and B-27 supplement (#17504–044, Invitrogen) for 2 weeks on poly-D-lysine (#P7280, Sigma-Aldrich)-coated cover-slides or cell culture dishes. For differentiation, SH-SY5Y cells were treated with 50 μM RA for 5 days.

## Infection with coxsackievirus B3 (CVB3)

The H3 variant of CVB3, the Woodruff strain, and EGFP-CVB3 were a kind gift from Dr. E. Jeon (Samsung Medical Center, Seoul, Korea). The virus was propagated in HeLa cells. Viral titers were determined using the plaque assay [77]. SH-SY5Y cells were incubated with CVB3 at the 1 multiplicity of infection (MOI) in serum-free DMEM for 1 h and further incubated in DMEM supplemented with 10% FBS for 30 h. For primary neurons, cells were incubated with the indicated MOI of CVB3 in neuron culture medium for 24 h. In the *in vivo* model, 8- to 11-week-old male mice were infected by intraperitoneal (IP) injection with $1.0 \times 10^6$ plaque forming units (PFUs) of CVB3 in 100 μl of phosphate-buffered saline (PBS).

## Tissue preparation

Mice were anesthetized and transcardially perfused first with perfusion solution containing 0.5% sodium nitrate and 10 U/ml heparin, and then with 4% paraformaldehyde in 0.1 M phosphate buffer (PB; pH 7.2). Brains were initially stored in 4% paraformaldehyde for 24 h at 4˚C, and then in a 30% sucrose solution until they sank. For reverse transcription polymerase chain reaction (RT-PCR) and western blotting, mice were perfused with the perfusion solution for 2 min and each organ was stored at −70˚C until use. For immunostaining, six separate series of 30 μm coronal brain sections were sectioned using a cryostat (model CM3050S, Leica, Wetzlar, Germany) and stored in an anti-freeze stock solution (PB containing 30% glycerol and 30% ethylene glycol, pH 7.2) at 4˚C before the experiments.

## Immunocytochemistry

Cells cultured on coverslips were washed three times with PBS and fixed with 4% paraformaldehyde for 10 min at room temperature. The fixed cells were washed several times with PBS and permeabilized with PBS containing 0.1% Triton X-100 for 5 min at room temperature. After washing with PBS, the cells were blocked with PBS containing 1% bovine serum albumin

(BSA) for 1 h at room temperature, and then incubated overnight with the indicated antibodies at 4˚C. The samples were then incubated with Alexa Fluor 488- (#A21202, #A21206) or Alexa Fluor 568- (#A10037, #A10042)-conjugated secondary antibodies (all from Invitrogen) for 1 h and then stained with to Hoechst for 5 min. The sections were mounted and observed by confocal microscopy (LSM710, Carl Zeiss, Jena, Germany) at the Three-Dimentional Immune System Imaging Core Facility of Ajou University. Live cells were incubated with 50 nM LysoTracker prepared in DMEM supplemented with 10% FBS for 30 min. After staining the nuclei with Hoechst for 5 min, live cells were observed by confocal microscopy.

## Western blotting

Samples were lysed using ice-cold RIPA buffer (50 mM Tris-HCl, pH 7.4, 0.5% sodium deoxycholate, 150 mM NaCl, 0.1% SDS, 1% Triton X-100) containing a protease inhibitor cocktail (#535140, Calbiochem, Darmstadt, Germany) and a phosphatase inhibitor cocktail (#P3200-001, GenDEPOT, Baker, TX). After brief sonication, the lysates were centrifuged at 16,000 x g for 30 min at 4˚C, and the supernatants were collected. In case of mice tissues (brains and hearts), after lysing and homogenizing the samples with TRIzol (#TR118, Molecular Research Center Inc, Cincinnati, OH, USA), proteins were isolated according to the manufacturer's protocol. The protein concentrations were determined using the DC Protein Assay Reagents Package (#5000116, Bio-Rad, Hercules, CA). Proteins were resolved by SDS-PAGE, transferred to PVDF or NC membrane, and immunoblotted with the indicated primary antibodies and subsequently with horseradish peroxidase-conjugated secondary antibody (#G-21040, Invitrogen or #111-035-003, Jackson ImmunoResearch, West Grove, PA). Proteins were then visualized using an enhanced chemiluminescence (ECL) system (#LF-QC0101, AbFrontier, Seoul, Korea). The band intensities were determined using ImageJ (NIH, Bethesda, MD).

## RT-PCR

Total RNA was isolated from samples using TRIzol (#TR118, Molecular Research Center Inc.) according to the manufacturer's protocol. Total RNA was reverse transcribed using AMV Reverse Transcriptase (#M0277L, New England Biolabs, Ipswich, MA). The transcript levels of target genes were quantified using 2 X KAPA SYBR Fast Master Mix (#kk4602, Kapa Biosystems, Cape Town, South Africa) using the StepOnePlus Real-Time PCR System (Applied Biosystems, Foster City, CA). For each target gene, the transcript level was normalized to that of GAPDH and was calculated using the standard ΔΔCT method. A complete list of *primer sequences is provided* in supplementary S1 Table.

## Transmission electron microscopy (TEM)

Control and CVB3 infected samples were fixed with 0.1M sodium cacodylate buffer (pH 7.4) containing 1% formaldehyde / 2% glutaraldehyde for 30 min at 4˚C [78]. The samples were rinsed twice with cold PBS, post-fixed with a mixture of 1% osmium tetroxide and 1% potassium ferricyanide, dehydrated in graded alcohol and embedded in Durcupan ACM resin (Fluka, Yongin, Korea). Ultrathin sections were obtained with the resin, mounted on copper grids, and counterstained with uranyl acetate and lead citrate. The specimens were observed using the Sigma 500 transmission electron microscope (Carl Zeiss, Jena, Germany) at the Three-Dimentional Immune System Imaging Core Facility of Ajou University.

## Mitochondrial morphology analysis

Mitochondrial morphology in TEM images was analyzed as described previously [33]. Based on the type of mitochondrial restructuring, four categories (type I-IV) were assigned. Type I comprised mitochondrial cristae that were regular, tightly packed, and longitudinally oriented. Type II comprised mitochondria with an abnormal shape or nonuniform size, and irregular cristae that lacked orientation and tightness. Type III comprised mitochondria with varied shapes and sizes, and discontinuous outer membrane, fragmented cristae, and swollen matrix. Type IV comprised mitochondria with a ruptured outer membrane, no cristae, and myelin-like transformation.

## Cytotoxicity assay

Cytotoxicity assays were performed using the LDH Cytotoxicity Assay Kit (#K311-400, BioVision; Mountain View, CA) according to the manufacturer's protocol. Following the CVB3 infection for 30 h, 50 µl of the medium and assay kit solution were mixed in the wells of an optically clear 96-well plate. Absorbance was measured within 10 min at 490 nm with an ELISA reader (Molecular Device, Wokingham, UK)

## Flow cytometry

Flow cytometry was performed as described previously [79]. After EGFP-CVB3 infection at the indicated time points, the cells were isolated and fixed by resuspension in 4% paraformaldehyde in PBS overnight. After washing with PBS and resuspension in 1% BSA in PBS, cells were analyzed using the FACS Aria III cell sorter (BD, Franklin Lakes, NJ) at the Three-Dimentional Immune System Imaging Core Facility of Ajou University. The data were analyzed using Flowing Software version 2.5.1 (Turku Bioscience, Turku, Finland).

## Immunohistochemistry (IHC)

Every serial section in each set was collected and washed with PBS containing 0.2% Triton X-100 (PBST). After blocking with 1% BSA in PBST, sections were incubated overnight at room temperature with the indicated primary antibodies. The samples were incubated with Alexa Fluor 488- (#A21202, #A21206) or Alexa Fluor 568- (#A10037, #A10042) conjugated secondary antibodies (all from Invitrogen) for 1 h and then stained using Hoechst for 10 min. After mounting on slides, the sections were treated with Sudan Black B solution (0.1% in 70% ethanol) to inhibit auto-fluorescence of mouse tissues. Images were captured using either a confocal microscope (LSM710, Carl Zeiss, Jena, Germanry) or a fluorescence microscope (Axioscan Z1, Carl Zeiss, Jena, Germanry) at the Three-Dimentional Immune System Imaging Core Facility of Ajou University.

## Cell density analysis

The hemispheres of mice were used for detecting VP1 by RT-PCR. The opposite hemispheres were used to analyze cell density. The hemispheres from three mice whose brain VP1 levels were close to the average were analyzed. We measured the number of cells (neurons) and the density of cells (microglia) per specific anatomical structure. Tuj-1 merged VP1 positive cells were counted from three equivalent locations for neurons and Iba-1 merged VP1 positive cells from six equivalent locations for microglia along the rostrocaudal axis. The cells were marked with the suggested color. Microglia were counted using the MetaMorph neurite outgrowth module (Molecular Devices). The density was obtained by dividing the counted value by the area of each anatomical structure. The anatomical structure was divided into cerebral cortex,

cerebral nuclei, fimbria, internal capsule, thalamus, hypothalamus, and midbrain. In the olfactory region, the cerebral cortex was divided into isocortex and olfactory areas, and the hippocampus was separated from the cerebral cortex in the section including the hippocampus. In the case of cerebral nuclei, the striatum and pallidum were largely divided, and the striatum was again divided into the dorsal region, lateral septa complex, and ventral region. The number of TH-positive cells in control and CVB3-infected mice brain hemispheres was counted manually in the immunohistochemistry images (Axioscan Z1, Carl Zeiss, Jena, Germany). The average number of SNpc sections for a mouse were 5 to 6 at 1:6 series in our experimental setting (30 μm sections). Numbers represent the total number of TH-positive cells, per mouse, multiplied by 6 to calculate the population estimate.

## Statistical analysis

All values of experimental data are expressed as mean ± SD. Statistical significance was evaluated using the unpaired t-test, one-way ANOVA, or two-way ANOVA using Graphpad Prism software (GraphPad, La Jolla, CA). Linear regression was performed to analyze the correlation between mCherry and EGFP signal using GraphPad Prism software.

## Supporting information

**S1 Text. Supplementary Materials and Methods.**
(DOCX)

**S1 Fig. H&E staining and cytotoxicity analysis between control and CVB3-infected dSH-SY5Y cells and investigation of α-syn expression between control and CVB3-infected samples, old control and PD patients.** (A) Immunocytochemistry (ICC) images of α-syn KO dSH-SY5Y cells infected with CVB3 (MOI 0.25) (red) for 24 h. Scale bar indicates 10 μm. (B) H&E staining of α-syn OE dSH-SY5Y cells of control and infected with CVB3 (MOI 0.25) for 24 h. Scale bar indicates 20 μm. Arrows indicate eosin positive inclusions of CVB3 infected cells. (C) Cytotoxicity analysis of WT and α-syn OE dSH-SY5Y cells infected with 0.25 MOI of CVB3 for 24 h, by the LDH assay. Values are derived from three independent experiments (n = 3), one-way ANOVA test with Tukey's multiple comparison test. (D) The relative expression levels of several mRNAs in control and CVB3-infected dSH-SY5Y cells (MOI 0.25) for 24 h [24–26]. Values are derived from three independent experiments (n = 3), unpaired t-test. (E) Gene expression omnibus (GEO) analysis of the relative levels of α-syn in control (n = 3) and CVB3-infected (intraperitoneal (IP) injection of 400 PFUs) mice (n = 3) hearts at day 4 posetinfection (PI) in 4 types of mice strains by array expression profiling (GSE19496). Differential gene expression (DGE) was analyzed using the Limma package in R. The P value of each analysis was as follows: A/J strain ($9.20E^{-07}$), B10.A-H2a Strain ($7.54E^{-06}$), B6.Chr 3A/J strain ($3.47E^{-05}$), CSS3 strain ($6.66E^{-09}$). (F) The relative levels of α-syn in postmortem brains of normal (n = 9) and patients with PD (n = 16) investigated by array expression profiling (GSE7621). DEG was analyzed using the limma package in R. P value = 0.002
(TIF)

**S2 Fig. Enterovirus 71 infection does not form autophagosomes colocalized with α-syn.**
ICC images of WT dSH-SY5Y cells either uninfected (control) or infected with enterovirus 71 (EV71) (MOI 1) for 24 h. Cells were immunostained with indicated antibodies. White arrows indicate LC3-positive aggregates.
(TIF)

**S3 Fig. Elevated levels of α-syn increases autophagy flux.** (A) Volcano plots of DEGs between control and α-syn increased environment. (The GEO data sets corresponding to

overexpression of human α-syn using a lentiviral vector in mouse midbrain neurons (GSE70368), induced pluripotent stem cells (iPSCs) of α-syn (SNCA) triplicated family (GSE30792) and the mouse striatum tissue of human α-syn transgenic (TG) mice (GSE116010) were analyzed). The light and dark gray dots indicate genes whose expression changed insignificantly and significantly, respectively. The red dots represent up-regulated genes and the blue dots represent down-regulated genes. GSE70368 and GSE116010 were analyzed using the Deseq2 package and GSE30792 was analyzed using the Limma package in R. (B) GO plot for "positive regulation of macroautophagy" (GO:0016239), "autophagosome maturation" (GO:0097352) and "lysosome organization" (GO:0007040) in gene ontology (GO) analysis of GSE70368, GSE30792 and GSE116010 which were higher than the controls in each set.
(TIF)

**S4 Fig. α-Syn expression in WT and TG mice brains, CVB3-infected neurons and microglia.** (A) Western blot was performed using control and α-syn TG mice brain lysates. Protein levels were quantified by densitometry. *** P < 0.001, unpaired t-test. IHC images of CVB3-infected mice brains at day 4, 7, and 28 PI (IP injection of $1.0 \times 10^6$ PFUs of CVB3). Tuj-1 (thalamus) (B) and Iba-1 (olfactory bulb) (C) positive cells, which were colocalized with VP1 in CVB3 infected WT and α-syn TG mice are shown. White arrows indicate CVB3 infected Tuj-1 positive cells. Blue indicates DAPI. Scale bar indicates 20 μm.
(TIF)

**S5 Fig. Activation of microglia by CVB3 infection in mice brains.** (A) ICC images of control and CVB3-infected mice brains at day 7 PI (IP injection of $1.0 \times 10^6$ PFUs of CVB3) and enlarged images with schematic diagram of microglia for analysis (black lined). Scale bar indicates 10 μm. (B) Morphology analysis of VP1-positive microglia in the brains of CVB3-infected mice at day 7 PI. The relative levels of cell body size, process length and span ratio between control and VP1-colocalized microglia. Values are derived from microglia randomly selected from the indicated anatomical structure of three mice. * P < 0.05, *** P < 0.001, unpaired t-test.
(TIF)

**S6 Fig. CVB3 induced cytotoxicity in mice brains.** ICC images of control and CVB3-infected mice brains at day 28 PI (IP injection of $1.0 \times 10^6$ PFUs of CVB3). White arrows indicate cleaved caspase-3 (c-caspase-3) positive cells. Scale bar indicates 10 μm. Blue indicates DAPI.
(TIF)

**S7 Fig. Tissue injury and replication of CVB3 in the organs of WT and α-syn TG mice.** (A) IHC images of Evans blue staining and intensity analysis of control (n = 3), CVB3-infected WT (n = 4), and α-syn TG mice (n = 6) hearts at day 7 PI (IP injection of $1.0 \times 10^6$ PFUs of CVB3). Scale bar indicates 500 μm. One-way ANOVA test with Tukey's multiple comparison test. (B) The relative levels of VP1 between WT (n = 4) and α-syn TG (n = 7) mice hearts at day 7 PI (IP injection of $1.0 \times 10^6$ PFUs of CVB3). Unpaired t-test was performed. (C) Hematoxylin and eosin staining of heart, pancreas, liver and spleen in control and CVB3-infected WT, and α-syn TG mice at day 7 PI (IP injection of $1.0 \times 10^6$ PFUs of CVB3). (D) Western blot was performed using lysates of brain, heart, and pancreas from control and α-syn TG mice.
(TIF)

**S1 Table. Primers for quantitative RT-PCR.**
(DOCX)

## Acknowledgments

We thank Dr. E. Jeon (Samsung Medical Center, Seoul, Korea) for kindly providing the H3 variant of CVB3, the Woodruff strain, and EGFP-CVB3. We thank Prof. Sun-Young Chang (College of Pharmacy, Ajou University, Suwon, Korea) for kindly providing EV71. We also thank NIFDS for providing C56BL/6-Tg (NSE-haSyn)Korl mice and associated information.

## Author Contributions

**Conceptualization:** Soo Jin Park, Uram Jin, Sang Myun Park.

**Data curation:** Soo Jin Park, Uram Jin.

**Formal analysis:** Soo Jin Park, Uram Jin.

**Funding acquisition:** Sang Myun Park.

**Investigation:** Soo Jin Park, Uram Jin.

**Methodology:** Soo Jin Park, Uram Jin.

**Project administration:** Soo Jin Park.

**Resources:** Soo Jin Park, Uram Jin.

**Software:** Soo Jin Park.

**Supervision:** Soo Jin Park, Sang Myun Park.

**Validation:** Soo Jin Park, Uram Jin.

**Visualization:** Soo Jin Park.

**Writing – original draft:** Soo Jin Park.

**Writing – review & editing:** Soo Jin Park, Sang Myun Park.

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
