## [Decision Letter · Decision Letter 0]

15 Mar 2021

Dear Dr. Park,

Thank you very much for submitting your manuscript "Interaction between coxsackievirus B3 infection and α-synuclein in Parkinson’s disease" for consideration at PLOS Pathogens. As with all papers reviewed by the journal, your manuscript was reviewed by members of the editorial board and by several independent reviewers. In light of the reviews (below this email), we would like to invite the resubmission of a significantly-revised version that takes into account the reviewers' comments.

Given the importance of microscopy data for the paper, please pay special attention to the image quality. Also, the reviewers raised important questions about the experiments in mice overexpressing a-synuclein. Better characterization of the murine model is required as well as controls allowing discrimination of pathological effects induced by a-synuclein overexpression without viral infection vs those that are virus-induced. 

We cannot make any decision about publication until we have seen the revised manuscript and your response to the reviewers' comments. Your revised manuscript is also likely to be sent to reviewers for further evaluation.

Sincerely,

George A. Belov, PhD

Associate Editor

PLOS Pathogens

Mark Heise

Section Editor

PLOS Pathogens

Kasturi Haldar

Editor-in-Chief

PLOS Pathogens

orcid.org/0000-0001-5065-158X

Michael Malim

Editor-in-Chief

PLOS Pathogens

orcid.org/0000-0002-7699-2064

Reviewer's Responses to Questions

**Part I - Summary**

Reviewer #1: This manuscript focuses on "interactions" between CVB3 and alpha-synuclein. There are a number of in vitro and in vivo experiments that the authors use to support this thesis.

This reviewer was concerned because there seems to be some blurring with respect to "association" vs. "causation/regulation". For example, it seems inappropriate for the authors to conclude on l. 292: "The present data demonstrate that CVB3 interacts with alpha-synuclein to promote PD."

Also, one wonders how different the results are compared to the effects of other picornaviruses.

An issue that this reviewer has with the manuscript is what looks like incomplete details regarding some the methods. For example:

a) l. 110 - it is unclear what the timing was when there was a decrease in alpha-synuclein mRNA, and whether other mRNAs were examined. Could this just be related to destruction of the cells?

b) l. 116 - alpha-synuclein protein increased in infected cells and decreased in neighboring cells. Isn't there translational shutoff of picornaviruses during infection? The ICC images may not be the best way to see this.

c) The authors use overexpressing alpha-synuclein cells and mice. A previous study by Masliah et al found that alpha-synuclein transgenic mice have neurodegeneration. Do the cells and mice used in this investigation also have abnormalities, and could these influence their results? In Fig. 6K, It would be valuable to see the survival rate or pathology noted for the uninfected alpha-synuclein transgenic mice.

d) The authors should provide virus titers in order to examine CVB3 replication.

e) l. 261 - Did the authors examine other tissues besides the heart and brain in the CVB3-infected mice?

f) Fig. 4H - At what time were the cells examined?

Reviewer #2: In this study, the authors hypothesize that coxsackievirus B3 (CVB3) can induce a-synclein (a-syn)-associated inclusion bodies in neurons suggesting this enterovirus can be a trigger for Parkinson’s Disease (PD). The authors demonstrate that in a neuronal cell line and in primary mouse neurons, that there is a correspondence of aa-syn and CVB3 VP1 in aggregates but that there is a decrease in a-syn mRNA expression and in brains of CVB3-infected mice, both a decrease in a-syn mRNA and protein in cells with CVB3 VP1 expression. The authors note that this is due to a neighboring cell inhibition of a-syn expression but not a normal innate response as poly I:C treatment does not have this effect. Is it possible that in the cell cultures at least, the loss of cells due to the CVB3 infection skews the a-syn expression level of the whole culture downward?

CVB3 is noted to form megaphagosomes containing disordered organelles and damaged mitochondria similar to the structures containing a-syn in Parkinson’s Disease (PD). CVB3 VP1 colocalizes with �-syn in aggregates in infected cells overexpressing a-syn. The authors hypothesize that as CVB3 is noted to block the late stage fusion of autophagosomes with lysosomes, the infection is a trigger for the generation of synucleinopathies leading to PD. The authors demonstrate an increase in CVB3 VP1 expression in cells and in mouse brains overexpressing a-syn. As a-syn overexpression itself has been shown to impair autophagic flux [Sarkar et al. (2021) PLOS Genetics 17(2): e1009359], it seems likely that the two processes affecting autophagy may result in an environment favoring generation of the necessary replication organelles for CVB3 replication and for generation of inclusion bodies.

The authors noted that although CVB3 VP1 did not co-localize with a-syn in the mice overexpressing a-syn, CVB3 infection did increase the level of pSer129 a-syn in western blots. Due to the difficulty in finding co-localization, the authors hypothesize the virus may act as a trigger rather than a continuing process of virus infection. Have the authors considered that evidence in human cardiomyopathy and in human type I diabetes combined with mouse models of these diseases suggests that enteroviruses may persist at extremely low levels but still cause pathology (Kim et al. J Virol. 2005 Jun;79(11):7024-41; Bouin et al. Circulation. 2019 May 14;139(20):2326-2338; Oikarinen et al. Viruses. 2020 Jul 11;12(7):747; Krogvold et al. Diabetes. 2015;64:1682–1687? It is extremely hard to detect the enterovirus by immunohistochemistry in these studies. In the brain in particular, the viral load is very low in adult mice, although infection of neonatal mice has been shown to induce persistence of the virus in the brain for 3 months post inoculation (Feuer et al J Virol. 2009;83(18):9356-9369.) In these studies, the use of RT-PCR allowed detection of the viral genome in infected tissue but only very sensitive detection of capsid protein demonstrated the presence of viral protein. The immunohistochemical method of detection of virus infection is inherently less sensitive than a method which allows amplification of the viral genome. This is not to say that the authors’ hypothesis is without merit, just that it cannot rule out the presence of the virus in the region where pathology occurs without doing an assay involving amplification of the low viral signal.

The observation that a-syn overexpression and CVB3 infection could result in Lewy body type inclusions in neuronal cells and that a-syn overexpression increased CVB3 replication does suggest the effects of both factors upon the induction of autophagy and the late stage autophagy block by CVB3 preventing clearance of aggregates may generate the type of pathology seen in PD. It is less certain that CVB3 and other enteroviruses may act as a trigger, rather than a persistent inducer of the inclusions as the authors have not done the type of assay noted for detecting the very low level chronic infection seen in other diseases such as cardiomyopathy and type 1 diabetes, specifically RT-PCR. As the authors have generated sufficient data to make this rather difficult process worthwhile in this animal model, the study merits publication but should discuss the limitations of immunohistochemistry in detection of the unusual persistence demonstrated with the coxsackievirus B viruses.

Reviewer #3: The authors claim a mutual interaction of CXB3 and a-syn where CXB3 is proliferating better in the presence of overexpressed a-syn, and conversely, the induction of aggregates of a-syn by CXB3, presumably in Lewy-body(LB)-like structures.

The topic, viral infections causing cellular pathology known from neurodegenerative diseases is interesting and timely and, as far as I could tell, coxsackie B3 (CXB3) has not investigated in the context of synucleinopathies apart from some case reports, so the paper is reporting novel findings that are principally worth to be published with potential consequences for future molecular, translational and clinical research in synucleinopathies.

The authors make a number of claims based on morphology but since the whole paper suffers from extremely poor image quality this is really hard to be confirmed by looking at those figures, even with a looking glass. In most cases microscopic images are less than 1 cm on a letter page meaning that in the final paper they will be even smaller. I propose that the authors present a novel manuscript with high-resolution images that will allow to better evaluate the claims they are putting forward.

**Part II – Major Issues: Key Experiments Required for Acceptance**

Reviewer #1: Please see Part I.

Titers of infections virus are necessary.

It would be valuable to carry out experiments in alpha-synuclein knock-out cells and mice.

It would be important to note the survival and pathology of alpha-synuclein transgenic mice.

Timing of data shown needs to be included.

Controls for some experiments are not shown. For example, the examination of other mRNAs or proteins; a comparison of CV3B with other picornaviruses.

Reviewer #2: (No Response)

Reviewer #3: In many cases, appropriate controls are missing. For example, it would be important to show differences in a-syn aggregates also in uninfected overexpressing SHSY cells (Fig 1). The authors repeatedly claim lewy body (LB)-like structures – they should not because they do not present appropriate evidence. For example PMID 32339655 shows should look in EM. Their TEM pictures in Fig. 2 are not revealing in the absence of immunogold-labeling for a-syn and CXB3 - if they could convincingly show fibrillar a-syn structures immunolabeled in TEM it would increase the paper’s impact. The animal studies are important and the finding that the tgSyn animals exhibit a higher mortality (I assume a normal life expectancy of this strain which is not reported) is interesting, As I said the asyn stainings claimed to be different in CXB3 infected vs. control suffer from poor quality and, in my opinion, are at this point inconclusive and not supporting the authors’ claims. There are many typos or grammatical mistakes. In the methods section I miss how the decrease of TH-positive neurons upon CXB3 infection (Fig. 6I) was quantified (stereology?).

**Part III – Minor Issues: Editorial and Data Presentation Modifications**

Reviewer #1: Please see Part I.

Reviewer #2: (No Response)

Reviewer #3: In summary, an interesting and timely study, but with major shortcomings and, as it stands,

unjustified conclusions that, eventually, could be addressed in a major revision when a much improved image quality is presented.

PLOS authors have the option to publish the peer review history of their article (what does this mean?). If published, this will include your full peer review and any attached files.

Reviewer #1: No

Reviewer #2: No

Reviewer #3: No
---

## [Decision Letter · Decision Letter 1]

9 Jul 2021

Dear Dr. Park,

Thank you very much for submitting your manuscript "Interaction between coxsackievirus B3 infection and α -synuclein in Parkinson’s disease" for consideration at PLOS Pathogens. As with all papers reviewed by the journal, your manuscript was reviewed by members of the editorial board and by several independent reviewers. In light of the reviews (below this email), we would like to invite the resubmission of a significantly-revised version that takes into account the reviewers' comments.

The reviewers appreciated your effrots but again raised major concerns if the essential claim of the paper that CVB3 infection induces a-synuclein condenstation is substantiated by the presented data, and warn against data overinterpretation. I agree with their comments and urge you to provide more compelling evidence of the phenomenon. You may consider correlative light-EM microscopy imaging to demonstrate the formation of a-synoclein aggregates in infected cells. Also, no plausible explanation of how expression of a-synuclein (or any other cellular protein) in CVB3-infected cells could be increased given the inhibiton of nuclear transcription, nucleo-cytoplasmic trafficking and capped mRNA translation in enterovirurs infected cells is provided. Experiments with high MOI  may help to differentiate the increase of expression and the condensation that may change the antibody reactivity. The possible cross-reactivity of the antibodies against cellular proteins with the viral proteins also raised reviewers' concern and should be resolved.

We cannot make any decision about publication until we have seen the revised manuscript and your response to the reviewers' comments. Your revised manuscript is also likely to be sent to reviewers for further evaluation.

Sincerely,

George A. Belov, PhD

Associate Editor

PLOS Pathogens

Mark Heise

Section Editor

PLOS Pathogens

Kasturi Haldar

Editor-in-Chief

PLOS Pathogens

orcid.org/0000-0001-5065-158X

Michael Malim

Editor-in-Chief

PLOS Pathogens

orcid.org/0000-0002-7699-2064

Reviewer's Responses to Questions

**Part I - Summary**

Reviewer #1: This manuscript claims to demonstrate that CVB3 infection induces alpha-syn inclusion in neurons that might acta as a trigger for PD. In CVB3-infected mouse brain, alpha-syn aggregates were present in the cell body of midbrain neurons. Alpha-synuclein overexpression increased CVB3 replication and cytotoxicity. Alpha-syn tgnic mice had decreased survival, enhanced survival and neuronal death. This reviewer had a difficult time reviewing this manuscript since the main figures were VERY difficult to read - despite a magnifying glass and Adobe Illustrator. The study is an interesting one, but the data at this time is not as convincing as this reviewer would like.

- The authors note that alpha-syn levels were decreased in the CVB3-infected mouse heart and brain of PD patients; one wonders whether this is the case because there is a change in cell types as well as cell death in the heart and brain. CVB3-infected cells growing in vitro had increased levels of alpha-syn compared to the bordering cells - is this because of interferon produced by the infected cells? Was the alpha-syn really increased OR was it just aggregated?

- This reviewer's impression is that LBs are eosinophilic inclusion bodies. Is this the case with CVB3 infection? If so, it would be best for the authors to note this.

- There was a statistical difference in the titer (l. 210) at 30 hours. One wonders what titer differences would be present at other time points. What would be helpful is if the authors put some of the P values within the body of the manuscript.

- The authors note that survival of the alpha-syn tgnic mice "was poorer" than that of WT mice. I'm not sure what the titer difference was since I could not see the figure well; however, this study should compare the transgenic mice with littermates that are not transgenic. The Methods section failed to madk it unclear whether this was always carried out.

- L. 285: "These results suggested..." It would be valuable if the data made the results clear.

- l. 292: Why do the authors think that there was no colocalization of alpha-syn and CVB3 VP1 at day 7. Shouldn't the virus cause an early infection? I was unable to see Fig. 7C well.

- L. 317: I don't think the authors have shown that CVB3 "promotes" PD. l. 444: "our findings clarify the mechanism of LB formation and the pathogenesis of PD associated with CVB3 infection." This reviewer would like to see more data. For example, it would be valuable if there was evidence of low levels of CVB3 genome by PCR in PD brains.

Reviewer #2: This study of the effects of alpha-synuclein (a-syn) expression and coxsackievirus B3 (CVB3) infection in neuronal cells is motivated by the possibility of the induction of synucleinopathies by enterovirus infection of cells in the brain. The authors have demonstrated in cell culture that there is co-localization of a-syn and CVB3 protein in cells. They have also demonstrated that overexpression of a-syn increases CVB3 virus yield in cell culture and that CVB3 infection decreases the endogenous expression of a-syn in cells neighboring infected cells. This surprising result is not due to simple innate immune response to infection as poly I:C increases levels of a-syn. The authors provide TEM data on the induction of megaphagosomes in neuronal cells and demonstrate that the megaphagosomes have some characteristics such as increased fibril length and damaged mitochondria found in Lewy bodies. It was telling that the a-syn overexpressing dSH-SY5Y and the wt dSH-SY5Y produced the characteristic viral arrays expected of a high virus yield which at least in the dSH-SY5Y cells is indicative of the viral presence in the absence of immuno-gold antibody labeling. The identification of virus outside of these arrays is not very definitive.

However, in the mouse model, based on C57Bl6 mice with a transgenic overexpression of a-syn, the mice did not have the colocalization of CVB3 protein with a-syn, despite the clear evidence of increased virus replication in the brain and the presence of a-syn aggregates in the transgenic mice with CVB3 expression. The authors were able to show with IHC the progression of the virus infection through the brain. As noted by the authors, the high mortality of the infected mice may have precluded the findings of synucleinopathies as clinical disease. C57BL/6 mice are not noted for long term CVB3 infections but are known to have high levels of acute pathology both in heart and pancreas. The clear evidence of pancreatic disease (both in pathology and body weight) induced by the high titer i.p. inoculum likely plays a part in the increased mortality. Is the more rapid mortality in the transgenic mice due to the increased a-syn expression in the brain alone? As a-syn has been shown to be expressed in islet beta cells (Steneberg, et al. 2015, Diabetes 62:2004–2014), it is possible that expression of a-syn at higher levels than normal in these beta cells could lead to increased susceptibility to CVB3 infection. Alteration of beta cell function could explain the increased mortality.

Although the mouse model left the question of the connection of enterovirus infection to synucleinopathies, the cellular data demonstrated an interaction with a-syn that increases viral replication and leads to increased formation of aggregates and fibrils in neurons. While this model does not provide proof of the role of CVB3 as a trigger of synucleinopathy, it does provide evidence of an interaction of this enterovirus B serotype with a-syn which may both provide a basis for further studies of models of Parkinson’s Disease and of diabetes.

Reviewer #3: I appreciate that the authors have attempted to improve the manuscript and certainly the images now offer a clearer view on their findings which allows me now for the first time to review these findings extensively.

Major issues: please see extra section.

In summary, after a thorough review of the data, I do not see that the authors’ general claim that CVB3 supports LB formation can be maintained. In fact, evidence from their in vivo experiments (Figure 7) or TEM studies (Figure 2) argues against it. What remains is that there is an effect on a-syn aggregation with immortalized cell lines. There are effects on a-syn expression regulation and a-syn seems to influence CVB3 replication but in vitro and in vivo findings are discrepant for the latter.

**Part II – Major Issues: Key Experiments Required for Acceptance**

Reviewer #1: The figures need to be easily readable.

The titers should be carried out at more than one time point.

Reviewer #2: None

Reviewer #3: Figure 1. why do the authors measure “relative intensity of a-syn“ rather than counting aggregates? The representative images 1G/K do not look convincingly different infected vs. non-infected. For ICC, the authors need to show a control that the anti-syn antibody does not crossreact with CVB3.

Figure 2. The authors state that they were not able to stain the fibril-like structures with a-syn immunogold antibodies. Contrary to the authors, I interpret the failure to see TEM immunogold labeling for a-syn as evidence that the fibril structures are NOT consisting of a-syn which makes a central claim of their paper fall apart. Furthermore the term LB-like is not suited for spherical structures such as the ones pointed to by arrows in Fig 2G/H since LBs have a concentric structure. Without having certainty about the identity of the observed fibril-like structures, the significance of the reported findings drops. The a-syn KO line results are only indirect and insufficient. The structures pointed at with the arrows in Figure 2 could thus be anything. In the absence of such a conclusive evidence, the authors should not continue to claim in several parts of the paper, including the abstract, that CVB3 causes a-syn aggregates. The sentences in the abstract “This study elucidated the mechanism of Lewy body formation and the pathogenesis of PD associated with CVB3 infection.“ or in the autor summary “Our findings clarify the mechanism of LB formation...“ are therefore wrong.

Figure 3A – the pS129 staining is too weak and does not show aggregates – the attempt of “staging“ therefore seems premature.

Fig 4A. the difference in virus titer measured by plaque assay is not even one order of magnitude, but only minimal. This could be thus entirely be due to a selection artifact when generating the stable overexpressing cell line. The plaque assay from Fig 6A is more convincing (two orders of magnitude titer difference) – but how do the authors then explain this discrepancy? As also mentioned by another reviewer, titer measurements in a-syn ko cells or mice should be performed to corroborate the claim of an interaction of a-syn with CVB3 replication

Figure 7: the staining of CVB3 and a-syn is too weak to support the authors’ claim that CVB3 infection supports a-syn aggregation in vivo, let alone LB formation.

**Part III – Minor Issues: Editorial and Data Presentation Modifications**

Reviewer #1: (No Response)

Reviewer #2: I suggest the authors provide a sentence or two discussing the role of pancreatic disease in the morbidity and mortality of the CVB3 infected mice (demonstrated by the pancreatic pathology and the body weight loss) and the possibility that a-syn expression in the pancreas leading to a significant increase in CVB3 replication in those tissues explains the higher morbidity and mortality in the a-syn OE transgenic mice. In addition, the crystalline arrays in the dSH-SY5Y cells should be in discussions of Figure 2E in the text.

Reviewer #3: (No Response)

PLOS authors have the option to publish the peer review history of their article (what does this mean?). If published, this will include your full peer review and any attached files.

Reviewer #1: No

Reviewer #2: **Yes: **Nora M Chapman, Ph.D.

Reviewer #3: No
---

## [Decision Letter · Decision Letter 2]

5 Oct 2021

Dear Dr. Park,

Thank you very much for submitting your manuscript "Interaction between coxsackievirus B3 infection and α -synuclein in Parkinson’s disease" for consideration at PLOS Pathogens. As with all papers reviewed by the journal, your manuscript was reviewed by members of the editorial board and by several independent reviewers. The reviewers appreciated the attention to an important topic. Based on the reviews, we are likely to accept this manuscript for publication, providing that you modify the manuscript according to the review recommendations.

In addition to addressing the final reviewers' comments it is important that you include the data you provided in the response letter as "figures for reviewers" in the manuscript. The readers of the paper should have access to all the material.  

Sincerely,

George A. Belov, PhD

Associate Editor

PLOS Pathogens

Mark Heise

Section Editor

PLOS Pathogens

Kasturi Haldar

Editor-in-Chief

PLOS Pathogens

orcid.org/0000-0001-5065-158X

Michael Malim

Editor-in-Chief

PLOS Pathogens

orcid.org/0000-0002-7699-2064

Reviewer Comments (if any, and for reference):

Reviewer's Responses to Questions

**Part I - Summary**

Reviewer #4: This is a very interesting manuscript regarding the effects of infection with CVB3 virus on alpha-synuclein aggregation in cultured neurons and in transgenic mice over expressing alpha-synuclein through the neuroenolase promoter. The paper addresses an important point in the field of virology and virally-induced neurodegeneration, namely the potential anti-viral or viral-response effects of alpha-synuclein, which are reported to be critically linked to innate immunity in the brain. The studies presented are thorough and the data compelling overall and deemed to add significantly to this field.

The data in figure 2 provide strong evidence that infection with CVB3 increases alpha-synuclein aggregation in SY5Y cells and in primary neurons. Staining for pS129 synuclein, combined with EM data, support that viral infection increases aggregation and/or fibril formation in neuronal soma, particularly in peri-mitochondria regions.

The authors have been very responsive to previous, including the provision of new data in several figures. As such, the revised manuscript is deemed largely suitable for publication.

One suggestion is to change the title to state in a “model of PD” or in a “mouse model of PD”.

**Part II – Major Issues: Key Experiments Required for Acceptance**

Reviewer #4: (No Response)

**Part III – Minor Issues: Editorial and Data Presentation Modifications**

Reviewer #3: I suggest the authors provide a sentence or two discussing the role of pancreatic disease in the morbidity and mortality of the CVB3 infected mice (demonstrated by the pancreatic pathology and the body weight loss) and the possibility that a-syn expression in the pancreas leading to a significant increase in CVB3 replication in those tissues explains the higher morbidity and mortality in the a-syn OE transgenic mice.  In addition, the crystalline arrays in the dSH-SY5Y cells should be in discussions of Figure 2E in the text.

Reviewer #4: One suggestion is to change the title to state in a “model of PD” or in a “mouse model of PD”.

PLOS authors have the option to publish the peer review history of their article (what does this mean?). If published, this will include your full peer review and any attached files.

Reviewer #4: No

Figure Files:

Data Requirements:

Reproducibility:

References:

---

## [Editor Report · Decision Letter 3]

8 Oct 2021

Dear Dr. Park,

We are pleased to inform you that your manuscript 'Interaction between coxsackievirus B3 infection and α-synuclein in models of Parkinson’s disease' has been provisionally accepted for publication in PLOS Pathogens.

Best regards,

George A. Belov, PhD

Associate Editor

PLOS Pathogens

Mark Heise

Section Editor

PLOS Pathogens

Kasturi Haldar

Editor-in-Chief

PLOS Pathogens

orcid.org/0000-0001-5065-158X

Michael Malim

Editor-in-Chief

PLOS Pathogens

orcid.org/0000-0002-7699-2064
---

## [Editor Report · Acceptance letter]

20 Oct 2021

Dear Dr. Park,

We are delighted to inform you that your manuscript, "Interaction between coxsackievirus B3 infection and α-synuclein in models of Parkinson’s disease," has been formally accepted for publication in PLOS Pathogens.

Best regards,

Kasturi Haldar

Editor-in-Chief

PLOS Pathogens

orcid.org/0000-0001-5065-158X

Michael Malim

Editor-in-Chief

PLOS Pathogens

orcid.org/0000-0002-7699-2064